# Disentangling listening effort and memory load beyond behavioural evidence: Pupillary response to listening effort during a concurrent memory task

Yue Zhang [1,2,3,4] *, Alexandre Lehmann [1,2,3,4], Mickael Deroche [1,2,3,4,5]

1 Department of Otolaryngology, McGill University, Montreal, Canada, 2 Centre for Research on Brain, Language and Music, Montreal, Canada, 3 Laboratory for Brain, Music and Sound Research, Montreal, Canada, 4 Centre for Interdisciplinary Research in Music Media and Technology, Montreal, Canada, 5 Department of Psychology, Concordia University, Montreal, Canada

☯ These authors contributed equally to this work.
* yue.zhang7@mail.mcgill.ca

**Data Availability Statement:** All relevant data are within the manuscript and its Supporting information files.

## Abstract

Recent research has demonstrated that pupillometry is a robust measure for quantifying listening effort. However, pupillary responses in listening situations where multiple cognitive functions are engaged and sustained over a period of time remain hard to interpret. This limits our conceptualisation and understanding of listening effort in realistic situations, because rarely in everyday life are people challenged by one task at a time. Therefore, the purpose of this experiment was to reveal the dynamics of listening effort in a sustained listening condition using a word repeat and recall task. Words were presented in quiet and speech-shaped noise at different signal-to-noise ratios (SNR): 0dB, 7dB, 14dB and quiet. Participants were presented with lists of 10 words, and required to repeat each word after its presentation. At the end of the list, participants either recalled as many words as possible or moved on to the next list. Simultaneously, their pupil dilation was recorded throughout the whole experiment. When only word repeating was required, peak pupil dilation (PPD) was bigger in 0dB versus other conditions; whereas when recall was required, PPD showed no difference among SNR levels and PPD in 0dB was smaller than repeat-only condition. Baseline pupil diameter and PPD followed different variation patterns across the 10 serial positions within a block for conditions requiring recall: baseline pupil diameter built up progressively and plateaued in the later positions (but shot up when listeners were recalling the previously heard words from memory); PPD decreased at a pace quicker than in repeat-only condition. The current findings demonstrate that additional cognitive load during a speech intelligibility task could disturb the well-established relation between pupillary response and listening effort. Both the magnitude and temporal pattern of task-evoked pupillary response differ greatly in complex listening conditions, urging for more listening effort studies in complex and realistic listening situations.

**Funding:** This work received funding from the MITACS Accelerate program, in collaboration with Oticon Medical Canada, under the grand number IT10517. (https://www.mitacs.ca/en/programs/accelerate; https://www.oticonmedical.com/ca). The funders had no role in study design, data collection and analysis, decision to publish, or preparation of the manuscript.

**Competing interests:** The authors have declared that no competing interests exist.

# 1 Introduction

Effortless as it seems, everyday communication is cognitively demanding. Degraded speech input induced by adverse listening conditions (e.g., background noise, reverberation etc.) and peripheral hearing loss introduces mismatch between perceived acoustic signals and their canonical forms [1–3]. Resolving this mismatch demands more resources from the finite pool of cognitive resources, leading to fewer resources for other cognitive tasks and eventually overload [4, 5]. Populations facing long-term auditory challenges are specifically at risk. For instance, people with hearing impairment and particularly those using cochlear implants (CI) often experience high and sustained effort, even when speech communication reaches satisfactory level [6–10]. CI listeners have to engage and deploy more cognitive resources to achieve a satisfactory level of speech communication due to electric hearing. Such elevated and sustained listening effort is associated with detrimental psychosocial consequences including greater need for recovery after work, increased incidence of sick leave and social interaction withdrawal [6, 11–13]. Therefore, there is a growing interest in the field of hearing science to conceptualise and quantify listening effort during speech perception for different populations.

Pupillometry (the continuous recording of changes in pupil diameter) has been one of most widely used methods for assessing listening effort. Its popularity can be attributed to its sensitivity to a wide range of cognitive tasks and processing that relate to the concept of listening effort [3, 14, 15]. Past studies have shown that pupil size varies with different speech intelligibility, hearing impairment, lexical manipulation, masker type, spectral resolution, memory load and divided/focused attention [4, 16–25]. Typically, when task demands increase, for instance, with lower SNR, degraded spectral resolution or more digits to remember, pupil size increases. However, when the task becomes so demanding that it exceeds the capacity limit, pupil size stops increasing and/or starts decreasing, forming a relation similar to inverse-U shape between task demands and listening effort [14, 26–32].

Because pupil size variation is the result of a complex interplay between the parasympathetic and sympathetic system, pupillometry can also reveal aspects of listening effort relating to fatigue, motivation and arousal [33–36]. For instance, Wang et al. [36] showed a negative correlation between the need for recovery and peak pupil dilation relative the baseline (PPD), supporting the assumption that high fatigue could be related to a reduced state of arousal (hence smaller pupillary response) [37]. Furthermore, pupillometry has a reasonable temporal locking to cognitive events, with some delay due to the slow locus coeruleus (LC)-norepinephrine (NE) response. Typically, the peak of event-evoked pupillary dilation arrives within the time window from 0.7 to 1.5 sec following the target stimuli [22, 38, 39]. This allows pupillary response to show trial-by-trial and within-trial variation in listening effort, which can reveal the underlying cognitive processing and allocation policy that are hardly measurable via behavioural outcomes. For instance, pupil size typically decreases with increasing trial/block numbers within one condition, suggesting fatigue or habituation with similar stimuli and task [17, 40–42]; it also varies with the level of engagement that changes from one trial to the next [43].

Due to these multiple influences on pupillary responses, there is only limited understanding of how pupil size varies in complex situations, where multiple cognitive functions are engaged and effort sustained over a period of time. Rarely in everyday life are people challenged by one task at a time. Even in a simple conversation, one needs to decode the incoming speech input embedded in various types of background noise, retain some information for mental processing, ponder over the best choices of words and articulate a verbal response, all of which require sustained cognitive processing over time. Understanding pupillary response to speech understanding in those situations is essential to conceptualise and quantify listening effort in ecological conditions, especially in the case of hearing aid or cochlear implant users.

Specifically, the relation between single task demand and pupil dilation has been shown and well-replicated in studies manipulating speech intelligibility and memory load [18, 26–31]. However, there are only a handful of pupillometry studies involving multiple and sustained tasks within hearing science. For instance, Karatekin et al. [16] found that pupil diameter increased progressively with more digits to remember during a digit span task and a dual task (digit span with visual response time task), but the rate of increase was shallower in the dual task than the single task. McGarrigle et al. [44] asked NH participants to listen to one short passage per trial, presented with multi-talker babble noise, and at the end of each passage judge whether images presented on the screen were mentioned in the previous passage. A steeper decrease in (baseline corrected) pupil size was found for difficult than easy SNR, but only in the second paragraph. This was interpreted as an index of the onset of fatigue in listening conditions requiring sustained effort. However, paragraphs were between 13-18s and the target word was periodically varied inside the paragraph, making it difficult to measure directly the pupillary response evoked by recognising and encoding the target item. In Zekveld et al. [45], participants had to recall the four-word cues (either related or unrelated to the following sentence) presented visually before the onset of the sentence embedded in a speech masker. The 7dB SNR difference between two sentence-in-noise conditions (-17dB and -10dB) elicited a difference in intelligibility, but not in peak and mean pupil dilation. This contrasted with the well-established effect of auditory task demands on the pupillary response, suggesting that an external cognitive load (i.e., memory) during speech processing could nullify the intelligibility effects on pupil dilation response. Overall, these studies point to a complicated, but under-investigated, relation between the speech task and the pupil dilation response, when other cognitive task load is present.

Therefore, the current experiment starts addressing the lack of systematic investigation on the dynamics of pupillary response in complex and sustained listening situations. To do so, we designed a behavioural paradigm including two TASKs with different demands in cognitive resources: a repeat-only condition where participants listen and repeat one word consecutively for ten words, and a repeat-with-recall condition where after listening and repeating each of the ten words, they need to recall as many words as possible at the end of the tenth word. Using words instead of digits or paragraphs, the paradigm utilises natural speech, yet still provides precise time-locking to the canonical task-evoked pupil response. The recall task poses a substantial and sustained requirement of cognitive resources (attention and working memory) that are also essential for speech understanding: participants had to complete both word recognition and memorising tasks within the same time window, and keep retaining more words in the memory until the end of the list. The task difficulty was further manipulated by embedding words in different levels of speech-shaped noise to compare pupillary responses under high and low listening effort (LISTENING condition). Simultaneously, pupil size variations were recorded. Participants' subjective ratings on effortfulness were also collected, and results were correlated with individuals' behavioural and pupillary responses. This analysis helps to disentangle further pupil responses corresponding to word recognition and memory, by identifying pupillary metrics that are significantly related to word recognition, recall and self-rating performance.

The main hypotheses were:

- Fewer words correctly repeated in difficult versus easy SNR conditions due to more degraded acoustic input, and fewer stated words recalled with more adverse SNRs due to limited cognitive capacity to prioritise the word recognition task.

- Bigger pupillary response in difficult versus easy SNR conditions, due to more degraded acoustic input. Bigger pupillary response in repeat-with-recall versus repeat-only condition:

bigger baseline pupil diameter due to accumulating memory load and bigger PPD due to greater cognitive demands. This difference might also depend on the serial position.

- Quick and large increase in pupil diameter when listeners were prompted to recall the words previously heard (similar to Cabestrero et al. [28]), and possibly bigger increase in difficult versus easy SNR conditions.

- Higher self-report effort in difficult SNR and repeat-with-recall conditions, reflecting the increased subjective experience of effort for conditions with more degraded acoustic input and sustained effort.

## 2 Methods and results

### 2.1 Methods: Participants

Data were collected from 25 adults (age range:18-49 years; average: 29 years; 17 females). A pure tone audiometry was administered to ensure that all participants had binaural thresholds at or better than 25 dB HL at 0.25, 0.5, 1, 2, 4, 8 kHz. All participants were native speakers of either French or North American English (10 French speakers). The study was always run in their native language. This work received ethical approvals from McGill University Faculty of Medicine Research Ethics Board (IRB) under the number A05-B11-18B. Prior to the experiment, participants were given enough time to read the information sheet and consent form approved by the ethics board. All gave written informed consent for their participation.

### 2.2 Methods: Stimuli

Stimuli were standard CNC words recorded from a male American English Speaker (mean duration = 0.61s, SD = 0.08s) and monosyllabic Fournier words [46] recorded from a male French Speaker (mean duration = 0.63s, SD = 0.11s). Words were fully randomised, grouped into lists of 10 and occurred only once in each list. Altogether, 240 words were used. They were then masked by speech-shaped noise (filtered on the long-term excitation pattern of the entire material, respectively in English or French) at three SNR levels of 0 dB, 7 dB and 14 dB. A quiet condition was also included, making a total of 4 LISTENING conditions.

Each LISTENING condition (three SNRs and quiet) was paired with TASK condition (repeat-only and repeat-with-recall) and was repeated three times (using three different word lists), making a total of 4x2x3 = 24 test blocks. Condition sequences and word lists were fully randomised for each participant.

### 2.3 Methods: Procedure

During the test, participants sat on a chair in a soundproof room, 2m in front of a 35-inch screen monitor and wearing an infrared binocular eyetracker (Tobii Glasses Pro2, 100 Hz sampling rate). The room and screen luminance levels were adjusted to reach 75lx (measured using a luxometer with the sensor positioned at the same height as the participants' left eye and facing the screen). The luminance levels were fixed throughout the experiment, to avoid changes in light level inducing task-unrelated pupillary response. Participants were instructed to minimise sudden movements that could dislocate the head-mounted eyetracker. The eyetracker was calibrated for each participant, and again during the experiment if participants made big postural changes. Pupil diameter (in mm), timestamps, horizontal and vertical gaze positions of the eyes were collected from the eyetracker outputs. All audio stimuli were presented through a Beyer Dynamics DT 990 Pro headphone via an external soundcard (Edirol

UA), calibrated at 65 dB SPL using a 1kHz pure tone. Experiments were run in Matlab 2016b, using Psychtoolbox and custom software.

After demonstrating the task and explaining the procedure, participants practised with one repeat-with-recall condition at 14dB SNR to familiarise themselves with the test sequence and requirements for pupil recording.

The SNR of each block was achieved by fixing the masker level and adjusting the target level. In the quiet condition, the speech level was set at 65 dB SPL. In this way, listeners could not estimate the upcoming block difficulty based on the noise level that was presented first (except for the quiet condition) [32]. The SNR of each trial within a 10-word block was held constant. Before each test block, participants were notified by words on the screen to either recall (printed in red) or not recall (printed in black) at the end of the ten words. Condition sequences were fully randomised for each participant. 3s after the notification, a black fixation cross appeared and stayed for another 1s, to indicate the start of the first trial and eliminate any carry-over effect from reading the coloured words in the pre-block notification. In each trial within the block, the presentation of speech-shaped noise masker (or quiet in the quiet condition) started 1.5s before the onset of the word. Participants were instructed to fixate on the black fixation cross displayed at the centre of the screen. After 1.5s, the word was played, and the presentation of the masker noise (or quiet in the quiet condition) was turned off 1s after the word offset. Upon the masker offset, the fixation cross turned into a circle, and this prompted participants to repeat back the word. They were instructed to fixate on the black circle during the verbal response. The experimenter then typed down the repeated word and pressed ENTER to proceed to the next trial. Words were scored automatically based on whether the characters typed matched the transcripts. The experimenter was always presented with the intended word on the Matlab interface, so potential homophones were scored as correct. No fixed time was enforced on the participants and experimenter to repeat back and type down the correct word. Both the participants and the experimenter were instructed to take time. This was to avoid extra mental stress and ensure the correct scoring of word recognition and recall performance. On average, it took 2.11s (SD = 1.08s) from the onset of the prompt cue to the onset of the next trial.

In blocks requiring recall, 2s after the end of the 10th trial, the word RECALL appeared on the screen followed by a black circle to prompt the participants to recall as many words as possible from the previous 10 words in any order. Participants were instructed to fixate on the black circle during recall. Their responses were typed down by the experimenter and scored automatically based on character matching with the response typed during word repeat. Therefore, correctly recalled words would include words that were correctly recalled misperceptions (similar to [47]), dissociating the impact of intelligibility from recall performance.

At the end of each block (containing a list of 10 words and recall session if it was a repeat-with-recall condition), participants were asked verbally to rate *How effortful the last block was* from 1 to 10, 10 being most effortful. Their subjective ratings were typed down by the experimenter. An illustration of the test sequence is shown in Fig 1.

The experiment lasted for 1 hour.

## 2.4 Behavioural data analysis and results

There were no differences between the French-speaking and English-speaking listeners in word recognition ($t = 0.32$, $df = 20.50$, $p = 0.75$), word recall ($t = 0.09$, $df = 20.66$, $p = 0.93$) and subjective rating ($t = 0.68$, $df = 22.57$, $p = 0.50$), using between-subjects Welch two-tailed t-tests. Therefore, data were firstly aggregated over language (as this played no role and was not a factor of interest in our study).

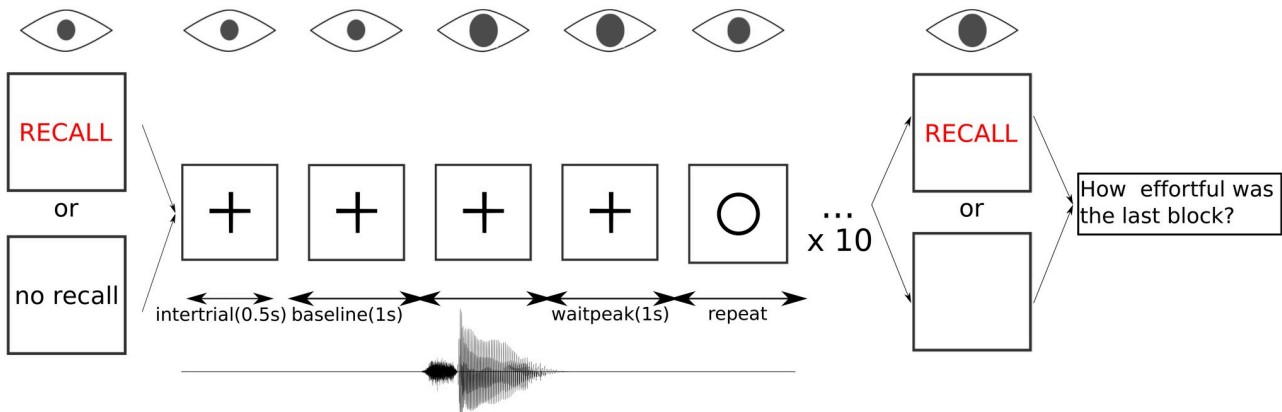

**Fig 1. Test sequence in a block.** Before each block, participants were presented with either words 'please listen, repeat and recall' in red or words 'please listen, repeat and no recall' in black against a white screen, indicating whether the incoming block was repeat-with-recall or repeat-only condition. 3s after the words notification, a black fixation cross appeared and stayed for another 1s, to signal the start of the first trial. The trial started with acoustic presentation of 0.5s speech-shaped noise (or quiet in the quiet condition) and visual presentation of a black fixation cross ('intertrial'). Another 1s of baseline measurement followed, with the same acoustic and visual presentation ('baseline'). The word was then played at 1.5s into the trial, followed by noise presentation (or quiet in the quiet condition) for 1s ('waitpeak'), with the same visual presentation. Upon the offset of 'waitpeak', the black fixation cross turned into a black circle to prompt listeners to repeat back the word 'repeat'. If the block was a repeat-with-recall condition, at the end of the 10th word, participants were prompted by the word RECALL followed by a black circle on the screen to start recalling previously repeated words. At the end of the block, participants were verbally reminded to rate *How effortful was the last block* from 1 to 10, 10 being most effortful.

**2.4.1 Methods: Word recognition performance.** To examine the effect of LISTENING and TASK conditions on word recognition, a logistic mixed-effect model was fitted on listeners' word recognition, using LISTENING and TASK conditions as fixed effect factors, with LISTENER and WORD LIST as random effect factors. Mixed effect models allow for controlling the variance associated with random factors without data aggregation. Therefore, by using LISTENER and WORD LIST used for stimuli as random effect factors in the model, we controlled for the variance in overall performance (random intercept) and dependency on other fixed factors (random slope) that were associated with LISTENER and WORD LIST. Models were constructed using the lme4 package [48] in R [49], and figures were produced using the ggplot2 package [50]. Fixed and random effect factors entered the model, and remained in the model only if they significantly improved the model fitting, using Chi-squared tests based on changes in deviance ($p < 0.05$). Differences between levels of each factor and interactions were examined with post-hoc Wald test. p values were estimated using the z distribution in the test as an approximation for the t distribution [51].

**2.4.2 Results: Word recognition performance.** There was a significant main effect of LISTENING condition ($\chi^2 = 684.11$, $df = 3$, $p < 0.001$) and interaction between LISTENING and TASK conditions($\chi^2 = 10.64$, $df = 3$, $p = 0.01$), but no main effect of TASK ($\chi^2 = 1.49$, $df = 1$, $p = 0.22$).

Post-hoc Wald test showed that word recognition performance at 0dB SNR condition (*mean* = 73.78%, *SD* = 10.37%) was the lowest of the four LISTENING conditions. 7dB SNR condition (*mean* = 94.4%, *SD* = 5.03%) had higher word recognition performance than 0dB SNR ($\beta = 1.8$, *se* = 0.13, $p < 0.001$), and lower performance than 14 dB SNR (*mean* = 97.57%, *SD* = 3.55%) ($\beta = -0.82$, *se* = 0.2, $p < 0.001$) and quiet condition (*mean* = 98.93%, *SD* = 2.07%) ($\beta = -1.92$, *se* = 0.34, $p < 0.001$). 14 dB SNR condition had lower performance than quiet condition ($\beta = -1.1$, *se* = 0.36, $p < 0.001$). At 0dB SNR, word recognition was higher in repeat-with-recall (*mean* = 76.27%, *SD* = 10.01%) than in repeat-only condition (*mean* = 71.73%, *SD* = 10.41%) ($\beta = 0.27$, *se* = 0.12, $p = 0.03$). Surprisingly, in quiet, word recognition was lower

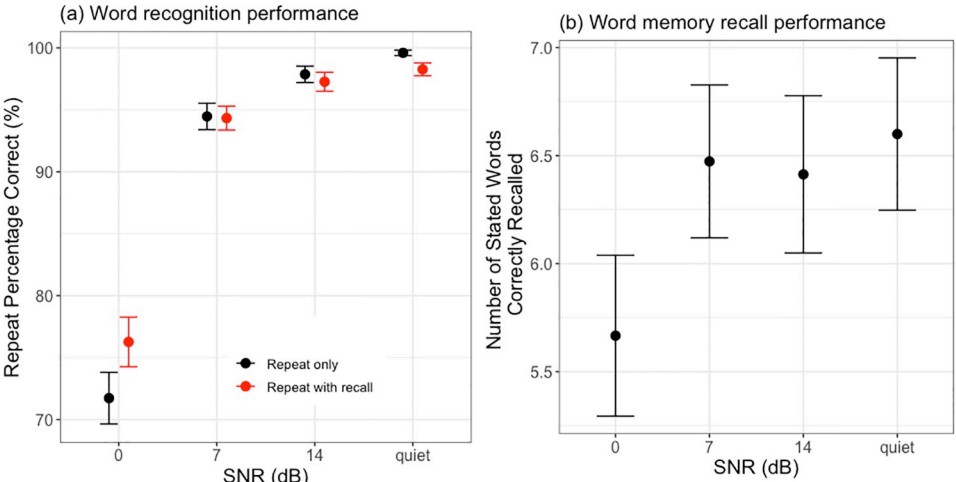

**Fig 2. Behavioural performance.** All data are averaged across 25 listeners. The error bars denote 1 standard error of the mean. (a) shows word recognition performance as a function of LISTENING and TASK conditions, and (b) shows free recall performance (when listeners were recalling as many words as possible the previously heard words from memory) as a function of the LISTENING condition.

in repeat-with-recall (*mean* = 98.26%, *SD* = 2.57%) than in repeat-only condition (*mean* = 99.6%, *SD* = 1.11%) ($\beta = -1.5$, *se* = 0.64, *p* = 0.02) (Fig 2). Recognition performance did not vary across ten word positions within each block ($\chi^2$ = 15.14, *df* = 9, *p* = 0.09).

**2.4.3 Methods: Word recall performance.**  To examine the effect of background noise on stated word recall performance, a logistic mixed-effect model was fitted on the number of words correctly recalled, with LISTENING condition as fixed effect factor, with LISTENER and WORD LIST as random effect factors, and following the same procedure reported above. Note that the recall performance was counted as stated word correct, and as such a word could be misunderstood and yet correctly recalled.

**2.4.4 Results: Word recall performance.**  There was a significant main effect of LISTENING condition ($\chi^2$ = 18.46, *df* = 3, *p* < 0.001). Post-hoc Wald test showed that fewer stated words were recalled at 0dB SNR (*mean* = 5.67, *SD* = 1.86) than 7dB SNR (*mean* = 6.47, *SD* = 1.77)($\beta$ = 0.38, *se* = 0.11, *p* < 0.001), 14dB SNR (*mean* = 6.41, *SD* = 1.82)($\beta$ = 0.34, *se* = 0.11, *p* = 0.003) and quiet condition (*mean* = 6.6, *SD* = 1.76)($\beta$ = 0.45, *se* = 0.11, *p* < 0.001), with no other significant differences (Fig 2b).

## 2.5 Methods: Pupil data preprocessing

Baseline pupil diameter in each trial was calculated as averaged pupil trace 1s before each word onset. The pupil diameter measured from the word onset to the end of the trial was subtracted from that baseline level to obtain relative changes in pupil diameter elicited by the task. Sample points were coded as blinks when pupil diameter values were below 3 standard deviation (SD) of the mean of the unprocessed trace or when gazing positions were 3 SD away from the centre of the fixation. Traces between 10 data points (0.1s) before the start and after the end of blink were interpolated cubically in Matlab, to further decrease the impact of the obscured pupil from blinks. Trials that had over 20% of the data points coded as blinks from the start of baseline to the start of the next trial were excluded. Trials containing blinks longer than 0.4s were also excluded, because they were more likely to be artefacts than normal blinks [52]. Three

participants had more than 20% of the overall trials discarded and were excluded from the pupillometry analysis (but kept for behavioural and subjective rating analysis).

All valid traces were low-pass filtered at 10 Hz with a first-order Butterworth filter to preserve only cognitively related pupil size modulation [53]. Processed traces were then aligned by the onset of the response prompt (the display of circle to signal participants to repeat back the word) and aggregated per listener, by each WORD POSITION in the 10-word list, TASK and LISTENING conditions.

## 2.6 Methods: Pupil data analysis

Two indices of task-evoked pupillary response (peak pupil dilation PPD and peak latency) were obtained from the aggregated traces, consistent with the method in [17]. PPD was the maximum diameter of pupil measurements from word onset to response prompt (time window 1), relative to the baseline pupil diameter. Note that we used the averaged pupil trace 1s before each word as the baseline during baseline correction, therefore, PPD corresponded to the phasic pupillary response evoked by word recognition. This method was in line with the aim of our experiment to investigate pupillary response to listening effort when another cognitive load was present. (For comparison, S1 Appendix. showed an alternative method to calculate PPD, i.e. baseline corrected by the averaged pupil trace 1s before the first word in the list, and its impact on understanding the results. To summarise, this alternative method could not disentangle the compound impact of listening effort and memory load on pupillary response.) Peak latency response was the time between word onset to the peak dilation. During this time window, listeners were predominantly listening and decoding the acoustic signals. There were also no significant differences in baseline pupil diameter ($t = 0.75$, $df = 19.7$, $p = 0.46$), PPD ($t = -0.49$, $df = 18.53$, $p = 0.63$) and peak latency ($t = 1.02$, $df = 17.04$, $p = 0.32$) between native English and French speakers using between-subjects Welch two-tailed t-tests, so data were aggregated over language.

**2.6.1 The effect of noise and memory load on pupillary response.** To investigate how the experimental manipulations on listening effort and memory load affected the dynamics of pupillary response, three mixed effect models were then fitted on baseline diameter, PPD and peak latency respectively. LISTENING and TASK conditions were entered as fixed effect factors to investigate the impact of experimental conditions on the pupillary response averaged over the ten-word list. WORD POSITION was coded as from 1 to 10, corresponding to the serial position of each word in the list. Entering this variable as another fixed factor enabled us to examine the temporal variations of different pupil metrics. Also, the interaction between WORD POSITION and other fixed effect factors showed how the pupil dynamics differed in the conditions with and without memory load, and under high and low listening effort. LISTENER was entered in the model as a random effect factor. Model buildings followed the same procedure above.

**2.6.2 Pupillary response of incorrectly versus correctly repeated words, recalled and forgotten words.** To further explore the sequence of different cognitive processing stages, pupil traces of words correctly versus incorrectly recognised, and pupil traces of words forgotten versus recalled were compared. For words correctly and incorrectly recognised, two logistic mixed effect models were fitted on the word recognition correct, using PPD and peak latency (calculated in time window 1 from word onset to response prompt) as fixed effect factors, with LISTENER as random effect factor. For words recalled and forgotten, a new time window was added into analysis. New PPD and peak latency were calculated at the time window from the response prompt to 1.5s after the response prompt (time window 2). The inclusion of extra 1.5s after the response prompt in the analysis was to include the time for rehearsing and

encoding the perceived word to working memory storage [54]. Logistic mixed effect models were fitted on the word recall, using PPD and peak latency in two time windows as fixed effect factors. Note that in this particular analysis pupillary parameters were used as independent variables to assess behavioural outcomes, to understand how the strategy of cognitive resources allocation affected word recognition and recall. In other words, it was examined as a predictive tool: predict whether a given word would be correctly understood or not, and recalled or forgotten, from the particular shape of a pupil trace.

**2.6.3 The effect of noise on pupillary response during word recall at the end of a block.** Finally, to explore the impact of LISTENING condition on the pupillary response (i.e. when listeners were recalling as many words as possible the previously heard words from memory), pupil traces from recall onset cue to 15s after the cue was firstly baseline-corrected by subtracting the average diameter of all previous word trials in the block. They were then de-blinked and low-pass filtered using the same parameters as above. Processed traces were then aggregated per listener by LISTENING condition. The mean of the trace during word recall was calculated. A mixed effect model was fitted on the mean pupil diameter during recall, with LISTENING condition as fixed effect factor and LISTENER as random effect factor.

## 2.7 Results: Pupil data

Figs 3a and 4a show the pupil diameter variation from the onset of baseline to 1.5s after the response cue.

**2.7.1 The effect of noise and memory load on pupillary response.** For baseline pupil diameter, there was a significant main effect of LISTENING condition ($\chi^2$ = 11.21, $df$ = 3, $p$ = 0.01), TASK ($\chi^2$ = 283.49, $df$ = 1, $p$ < 0.001) and WORD POSITION ($\chi^2$ = 24.85, $df$ = 9, $p$ = 0.003), and significant interaction between TASK:WORD POSITION ($\chi^2$ = 82.99, $df$ = 9, $p$ < 0.001). Post-hoc tests showed that baseline pupil diameter at 0dB SNR (*mean* = 3.89, *SD* = 0.76) was not different from 7dB SNR condition (*mean* = 3.91, *SD* = 0.81) ($\beta$ = 0.004, *se* = 0.01, $p$ = 0.68). Both were bigger than 14dB SNR condition (*mean* = 3.84, *SD* = 0.74) ($\beta$ = 0.04, *se* = 0.01, $p$ = 0.002; $\beta$ = 0.03, *se* = 0.01, $p$ = 0.007) and quiet condition (*mean* = 3.86, *SD* = 0.78) ($\beta$ = 0.04, *se* = 0.01, $p$ = 0.04; $\beta$ = 0.03, *se* = 0.01, $p$ = 0.04); 14dB was not different

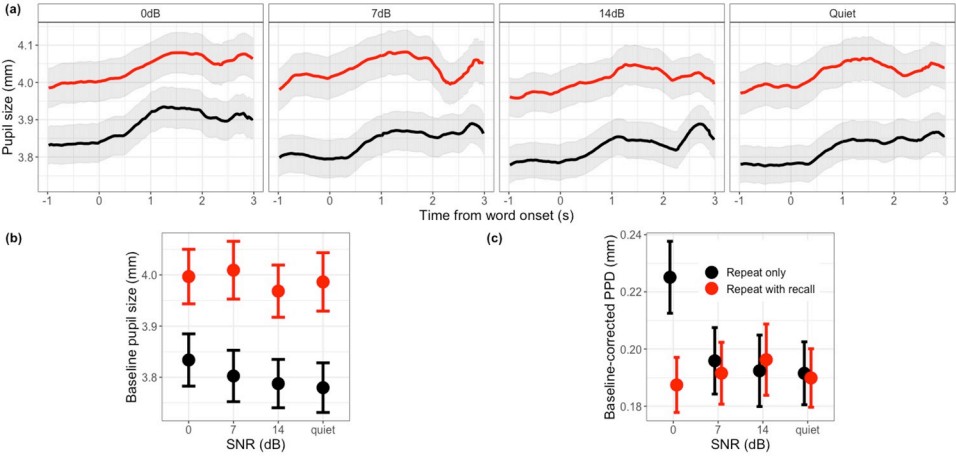

**Fig 3. Pupillometry results as a function of LISTENING and TASK conditions.** All data are aggregated across 22 listeners, and WORD POSITION, LISTENING, TASK conditions. The error bars and shaded width denote 1 standard error of the mean. (a) shows changes in pupil size as a function of time during each trial, for each LISTENING and TASK conditions. (b) and (c) plot baseline pupil diameter and PPD as a function of LISTENING and TASK conditions respectively.

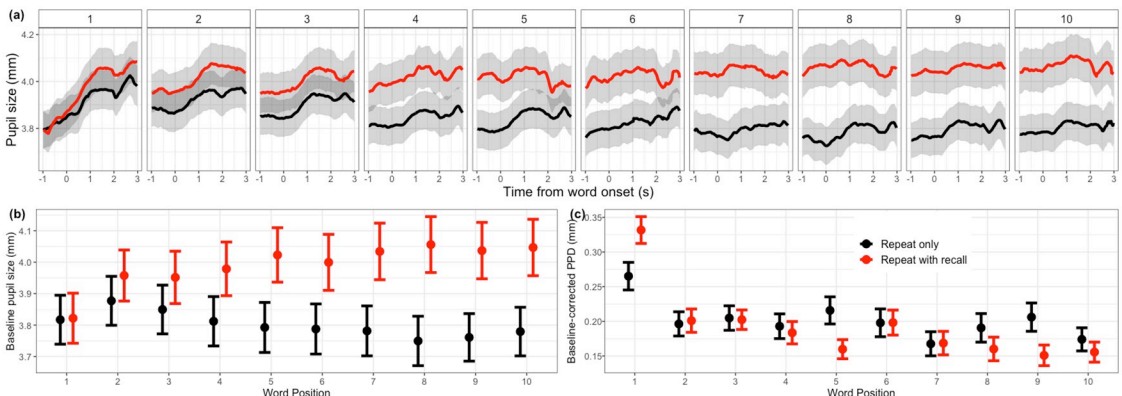

**Fig 4. Pupillometry results as a function of TASK and WORD POSITION.** All data are aggregated across 22 listeners, and WORD POSITION, LISTENING, TASK conditions. The error bars and shaded width denote 1 standard error of the mean. (a) shows changes in pupil size as a function of time at each WORD POSITION for each TASK condition. (b) and (c) plot baseline pupil diameter and PPD as a function of WORD POSITION and TASK condition respectively.

from quiet ($\beta = 0.01$, $se = 0.01$, $p = 0.32$). Overall, baseline pupil diameter at repeat-with-recall condition (*mean* = 3.95, *SD* = 0.78) was significantly bigger (about 0.15 mm) than that at repeat-only condition (*mean* = 3.81, *SD* = 0.76) ($\beta = 0.18$, $se = 0.01$, $p < 0.001$) (Fig 3b). A trend analysis on WORD POSITION showed that from the 1st to 10th word, repeat-only condition had a linearly decreasing trend ($\beta = -0.18$, $se = 0.01$, $p < 0.001$), whereas repeat-with-recall condition had a linearly increasing trend ($\beta = 0.18$, $se = 0.01$, $p < 0.001$) (Fig 4b). Baseline diameter in repeat-with-recall condition also showed a significant quadratic trend ($\beta = -0.09$, $se = 0.03$, $p < 0.001$), suggesting that the greatest increase in baseline diameter occurred in the mid-section of the word list. No significant cubic trend was detected.

For PPD, there was a significant main effect of WORD POSITION ($\chi^2 = 104.39$, $df = 9$, $p < 0.001$), and no significant main effect of LISTENING ($\chi^2 = 2.55$, $df = 3$, $p = 0.47$) and TASK conditions ($\chi^2 = 1.85$, $df = 1$, $p = 0.17$). Interactions between LISTENING:TASK ($\chi^2 = 13.15$, $df = 3$, $p = 0.004$) and TASK:WORD POSITION ($\chi^2 = 22.98$, $df = 9$, $p = 0.006$) were significant, with no significant three-way interaction ($\chi^2 = 31.05$, $df = 27$, $p = 0.27$). Post-hoc tests showed that at 0dB SNR, repeat-only condition (*mean* = 0.23, *SD* = 0.19) evoked bigger PPD than repeat-with-recall condition (*mean* = 0.19, *SD* = 0.14) ($\beta = 0.03$, $se = 0.01$, $p = 0.04$), and no difference between two tasks at other SNR levels (Fig 3c). Examining the same interaction differently: SNR only affected the repeat-only condition, showing a bigger PPD at 0 dB than at other SNR conditions. A trend analysis on WORD POSITION showed that from the 1st to the 10th word, there was a decrease in PPD ($\chi^2 = 55.73$, $df = 1$, $p < 0.001$, $\beta = -0.08$, $se = 0.01$, $p < 0.001$), and this decrease was steeper in the repeat-with-recall condition than repeat-only condition ($\beta = -0.07$, $se = 0.007$, $p < 0.001$) (Fig 4c). No further significant quadratic or cubic trend.

For peak latency, there was a significant main effect of LISTENING condition ($\chi^2 = 8.67$, $df = 3$, $p = 0.03$) and WORD POSITION ($\chi^2 = 66.98$, $df = 9$, $p < 0.001$), and significant interaction between TASK:WORD POSITION($\chi^2 = 21.93$, $df = 9$, $p = 0.009$). Post-hoc test showed that at 0dB SNR condition (*mean* = 1.12, *SD* = 0.59) pupil size peaked significantly later than at 7dB SNR (*mean* = 1.06, *SD* = 0.61) ($\beta = 0.07$, $se = 0.03$, $p = 0.008$), 14dB SNR (*mean* = 1.05, *SD* = 0.61) ($\beta = 0.06$, $se = 0.02$, $p = 0.01$), and quiet (*mean* = 1.06, *SD* = 0.59) ($\beta = 0.05$, $se = 0.03$, $p = 0.05$). From the 1st to the 10th word, there was an increase in repeat-only condition ($\beta = -0.11$, $se = 0.04$, $p = 0.007$), and also an increase ($\beta = -0.3$, $se = 0.04$, $p < 0.001$) in

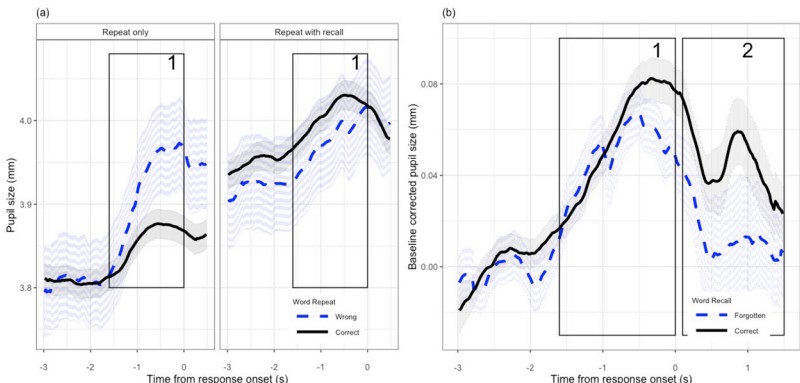

**Fig 5. Comparing pupil traces for words correctly and incorrectly repeated, recalled and forgotten.** All data are averaged across 22 listeners. The shaded width denotes 1 standard error of the mean. (a) compares the pupil traces for words correctly and incorrectly repeated in each TASK condition. (b) compares the pupil traces for words that are successfully recalled or forgotten. Traces in two time windows are analysed: first analysis window is from the onset of word to the onset of the response prompt, and the second analysis window is from the onset of the response prompt to 1.5s after the prompt.

repeat-with-recall condition, but steeper than repeat-only condition ($\beta$ = 0.2, $se$ = 0.05, $p$ = 0.001). No further significant quadratic or cubic trend.

**2.7.2 Pupillary response of incorrectly versus correctly repeated words.** For the pupillary responses of words that were correctly and incorrectly recognised, no difference in baseline diameter was found ($\chi^2$ = 0.001, $df$ = 1, $p$ = 0.94), suggesting that there was no differential arousal that could explain the word intelligibility. There was a main effect of PPD ($\chi^2$ = 12.59, $df$ = 1, $p < 0.001$) and a significant interaction of TASK:PPD ($\chi^2$ = 13.9, $df$ = 1, $p < 0.001$). No significant effect of peak latency ($\chi^2$ = 1.96, $df$ = 1, $p$ = 0.16) was found. Post-hoc tests showed that at repeat-only condition, bigger PPD was associated with incorrectly repeated words ($\beta$ = $-1.8$, $se$ = 0.35, $p < 0.001$), and no such relation at repeat-with-recall task (Fig 5a).

**2.7.3 Pupillary response of recalled versus forgotten words.** Comparing the pupillary responses of words that were later recalled or forgotten, no difference in baseline size was found ($\chi^2$ = 0.001, $df$ = 1, $p$ = 0.9). At the first time window, there was no significant main effect of PPD ($\chi^2$ = 1.76, $df$ = 1, $p$ = 0.18) and latency ($\chi^2$ = 1.49, $df$ = 1, $p$ = 0.22). At the second time window, there was a significant main effect of peak pupil diameter ($\chi^2$ = 4.87, $df$ = 1, $p$ = 0.03). Post-hoc Wald test showed that bigger PPD at the second time window was associated with the successful recall of the word ($\beta$ = 3.18, $se$ = 1.47, $p$ = 0.03) (Fig 5b).

**2.7.4 The effect of noise on pupillary response during word recall at the end of a block.** For the mean pupil diameter during the listeners' word recall, there was no difference among SNRs ($\chi^2$ = 0.67, $df$ = 3, $p$ = 0.88) (Fig 6); and the mean pupil diameter jumped from about 4.0mm to 4.3-4.4 mm (just short of 10%).

## 2.8 Methods: Subjective listening effort rating and individual differences

To examine the effect of LISTENING and TASK conditions on subjective rating, a logistic mixed-effect model was fitted on ratings, with LISTENING and TASK conditions as fixed effect factors, with LISTENER and WORD LIST as random effect factors, and following the same procedure reported above.

In a final attempt to delineate different components of the pupillary dynamics, each participant's pupillary responses (baseline diameter and PPD) were correlated with their age, word recognition, word recall and subjective rating performance using Pearson correlation.

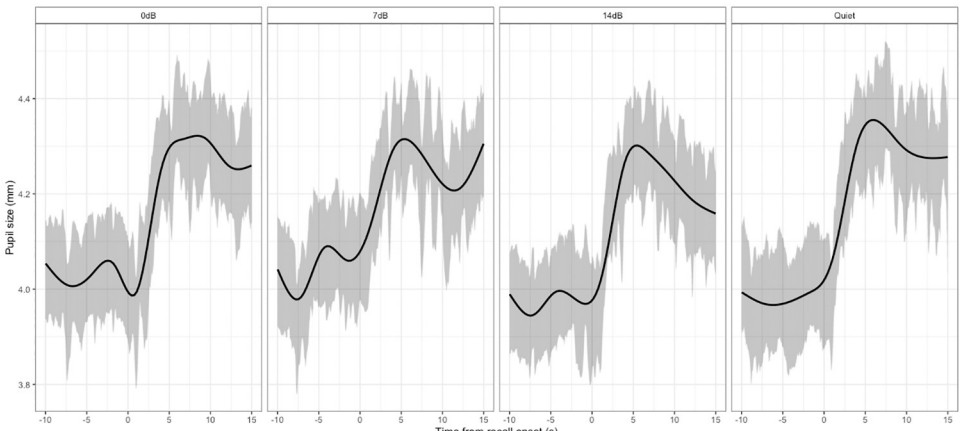

**Fig 6. Pupil traces from 10s before the recall onset to 15s after the recall onset.** Each panel shows the averaged traces in each LISTENING condition. All data are aggregated across 22 listeners. The shaded width denotes 1 standard error of the mean. The line is further smoothed using the default gam method in ggplot2 package to highlight the general trend.

All best fitting models, model parameter estimates and model comparison statistics were reported in the S2 Appendix.

## 2.9 Results: Subjective listening effort rating

There was a significant main effect of LISTENING ($\chi^2$ = 2278.51, $df$ = 3, $p < 0.001$) and TASK conditions ($\chi^2$ = 7137.01, $df$ = 1, $p < 0.001$), and a significant interaction of LISTENING: TASK ($\chi^2$ = 239.78, $df$ = 3, $p < 0.001$) on subjective rating. Subjective rating at 0dB (*mean* = 5.87, *SD* = 2.13) was higher than at 7dB (*mean* = 4.05, *SD* = 2.29) ($\beta$ = 0.85, *se* = 0.04, $p < 0.001$), 14dB (*mean* = 3.98, *SD* = 2.43) ($\beta$ = 0.89, *se* = 0.04, $p < 0.001$) and quiet (*mean* = 3.24, *SD* = 2.36) ($\beta$ = 1.29, *se* = 0.05, $p < 0.001$); 7dB was higher than quiet ($\beta$ = 0.44, *se* = 0.05, $p < 0.001$) but not 14dB ($\beta$ = 0.04, *se* = 0.05, $p$ = 0.38); and 14dB was higher than quiet ($\beta$ = 0.4, *se* = 0.05, $p < 0.001$). Overall, subjective rating at repeat-with-recall condition was higher than that at repeat-only condition ($\beta$ = 1.56, *se* = 0.03, $p < 0.001$), and the difference was smaller at 0dB than other SNR levels ($\beta$ = −1.13, *se* = 0.06, $p < 0.001$) (Fig 7a).

## 2.10 Results: Individual differences

On an individual level, baseline diameter (within word lists) positively correlated with word recall performance ($r$ = 0.45, $p$ = 0.04, Fig 7b), and negatively correlated with subjective rating ($r$ = −0.45, $p$ = 0.04, Fig 7c). PPD negatively correlated with word recognition performance ($r$ = −0.48, $p$ = 0.02, Fig 7d), but this was only true when no memory requirement was involved: in repeat-with-recall condition, there was no significant correlation between PPD and word recognition performance ($r$ = 0.08, $p$ = 0.21). Note that these relations were modulated by participants' age: word recall performance worsened with age ($r$ = −0.5, $p$ = 0.01); baseline diameter shrunk with age ($r$ = −0.52, $p$ = 0.01); and subjective rating shifted up with age ($r$ = 0.5, $p$ = 0.01). After correcting for the effect of age, the correlations were not significant between baseline diameter and word recall performance ($r$ = 0.18, $p$ = 0.08), and between baseline diameter and subjective rating ($r$ = −0.21, $p$ = 0.22). PPD and word recognition performance ($r$ = −0.02, $p$ = 0.01) remained significant after the correction.

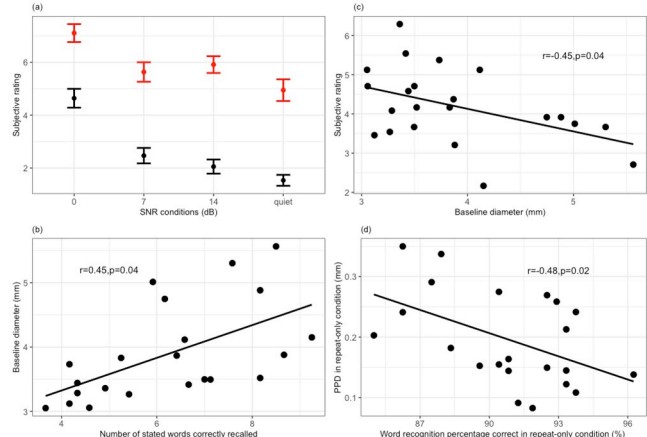

**Fig 7. Subjective effort and individual differences.** Each data point corresponds to one participant. The error bars denote 1 standard error of the mean. (a) plots subjective rating as a function of LISTENING and TASK conditions. (b) to (d) show the significant correlations ($p < 0.05$) between behavioural and pupillary measures.

## 3 Discussion

The current experiment used a word recall paradigm to elicit sustained and concurrent memory load on word recognition in noise. Pupil diameters were recorded simultaneously to investigate the dynamics of pupillary response in complex listening situations. A number of our findings can be contrasted with the literature, advancing current debates on 1) interferences between concurrent tasks, 2) the nature of pupil dynamics in dual versus single tasks, 3) the predictive power of pupillometry for intelligibility and memory, and 4) individual differences.

### 3.1 Word recall task interfering with the word recognition task

Consistent with our first hypothesis, results showed that noise impaired both word recognition and recall. Fewer stated words were recalled at 0dB than 7dB, 14dB and quiet conditions. Note that to dissociate the impact of word recognition from recall performance, word recall scoring was based on whether the recalled words matched the words repeated by participants, rather than the transcripts (similar to [47]). Past studies using the recall paradigm reported similar results. For instance, Surprenant showed that even when nonsense syllable recognition performance was similar across SNRs, NH participants' recall performance was impaired in difficult SNR [55]. In [56], NH participants repeated the final word of each of 8 sentences embedded in babble-speech noise, and at the end of the 8th sentence recalled as many of the previously reported words as possible. Results showed that challenging signal-to-noise (SNR) condition impaired both word recognition and recall of the stated words performance. Ng et al. [57] tested moderate to severe hearing loss participants using a similar memory recall paradigm referred to as the sentence-final word identification and recall (SWIR). Results showed that even under similar intelligibility, babble-speech noise impaired word recall performance more than speech-shaped noise. Similar effect was also illustrated for young versus old listeners [58], native versus foreign speech masker [59] and replicated in different languages [47]. In line with the interpretation in previous studies, we believe that this SNR effect on recall reflects that higher listening effort during word recognition evoked at lower SNR leaves fewer cognitive resources for encoding and retrieving words, leading to the decreased performance in the word recall task [4, 8, 60–62].

Surprisingly, we found a possible interference from the recall task on the word recognition task. At 0dB, word recognition performance was better when participants expected word recall at the end of the list; and in quiet, word recognition was worse when participants expected word recall task at the end. Although word recognition was essentially the same task in repeat-only condition and repeat-with-recall condition, participants might evaluate and anticipate the amount of cognitive resources differently. At 0dB, listeners might be more attentive and ready to engage overall more cognitive resources when they were notified at the beginning of the block that they should recall at the end of 10th word because they anticipated the incoming block to be demanding. When no recall was required, they might have judged beforehand that the incoming block was not worthwhile to mobilise too many resources, hence worse recognition performance. If this were the case, then we should observe a corresponding interaction in baseline pupil diameter, because baseline has been shown to be associated with task readiness and engagement [33, 43]. Trend analysis from the 1st to the 10th word showed that the baseline diameter in the repeat-only condition had a decreasing trend that was consistent with other studies where listener showed fatigue or habituation with similar stimuli and task within a block [17, 40–42] In comparison, the baseline diameter from the 1st to the 10th word in the repeat-with-recall condition had an increasing trend. However, without manipulation of both memory load and task engagement in our experimental design, it is impossible to disentangle the convoluted effect of memory accumulation and engagement on the baseline. Furthermore, in quiet with repeat-with-recall condition, listeners should have sufficient capacity to reach a better primary task performance (as shown by a higher word recognition in repeat-only condition), but instead, they performed worse in the word recognition task compared to in the repeat-only condition. This might suggest that they did not prioritise the word recognition task (although they were instructed explicitly to do so by the experimenter), and may have shifted some resources to the recall task probably because it was more interesting and rewarding [63–66].

This interference warrants further investigation, because it concerns the validity of using a dual-task paradigm in measuring listening effort. In order to interpret safely the difference in secondary task performance as a result of listening effort, implicit assumptions of the dual-task paradigm need to be reviewed [67]. Firstly, the paradigm assumes that participants have a limited pool of cognitive resources, but the Framework for Understanding Effortful Listening (FUEL) model also notes that resources that are available to be allocated are fluctuating with other factors besides overall task demands [3, 4]. In other words, the relationship between task difficulty and effort is not linear, but modulated by factors like fatigue, motivation and (dis) pleasure [35, 68–73]. Secondly, the paradigm assumes that listeners, under explicit instructions, will prioritise the primary task by investing as many resources as possible, and only leaves whatever left of the resources for the secondary task. However, individual differences and task characteristics might affect listeners' actual strategy [3]. For instance, older adults may differ from younger adults in the extent to which they prioritise one task over another [63–65]. And when the primary task is too complex or secondary task more novel, participants may consciously or unconsciously shift more resources to the secondary task relative to the primary task [74–76]. Although various recall paradigms from previous studies are sensitive to the relative allocation of cognitive resources [47, 56, 57, 59], there is no direct method to gauge the total amount of resources deployed and how they are allocated [67]. As illustrated in the current experiment, listeners might not mobilise and/or allocate the same amount of cognitive resources for the speech recognition task when a secondary recall task was anticipated, even under explicit instruction. This makes it unclear whether the difference in the recall performance is due to differences in the listening effort, or prior mobilisation of overall cognitive resources, or internal shift of resources between primary and secondary task. Previous studies

using the SWIR paradigm have typically fixed the SNR levels at or close to ceiling performance, to ensure no substantial differences in sentence intelligibility [47, 57]. But this still does not exclude the possibilities mentioned above, because even at ceiling performance level (similar to the quiet condition in the current experiment), interferences could occur.

Our results highlight the importance of considering these factors when designing behavioural paradigms for measuring listening effort and interpreting their outputs. The possible interference from the recall task on the word recognition task showed that the behavioural outcome might not be indicative of the listening effort alone. This might be particularly important when applying the test to listener groups who are susceptible to fatigue and task interference, for instance hearing impaired populations and children, because they might either give up or not fully be motivated in the first place even when the available capacity can meet the processing demand [3, 72, 74–76].

## 3.2 Pupillary response to intelligibility during a concurrent and sustained memory load

Consistent with our second hypothesis regarding the pupillary response during the word listening and encoding section, pupil diameter was larger in repeat-with-recall than repeat-only condition. In this respect, the present design has the advantage of dissecting how this difference arises, thanks to the trial-by-trial sensitivity of pupillometry. The difference arises from a progressive decrease in pupil diameter within the repeat-only condition, and a progressive increase in baseline diameter within the repeat-with-recall condition from the 1st to the 10th word. Although past studies have reported similar trends, they were using different materials and test designs, making it hard to demonstrate clearly the impact of additional memory task on listening effort in both magnitude and dynamics. For instance, within one speech perception task, pupil diameter gradually decreased with increasing trial numbers, due to task/stimuli habituation [17, 40–42]. However, when listeners needed to remember the digits [16, 26, 28] or pseudo-words [77] presented auditorily, pupil diameter increased progressively, until the memory span was exceeded. Note that in the current experiment, listeners needed to continuously decode words embedded in noise, which might be more effortful than listening to digits or pseudo-words in quiet due to higher cognitive and perceptual processing demands. The more demanding primary speech recognition task led to more accumulated and sustained effort over time. This might explain earlier plateau in baseline diameter in our experiment than observed in those studies. We observed a quadratic trend of baseline pupil diameter from the 1st to the 10th word within a list. [28] reported the plateau at the 9th digit for young adults, and [78] reported the plateau at 6th digits for children and 8th digit for adults. Our results are in good agreement with such estimates, and confirm that additional memory task places a heavier and sustained load on cognitive effort. More specifically, baseline diameter could reveal the impact on cognitive effort from the additional task, and the rate of increase in baseline diameter could be suggestive of the magnitude of sustained effort in a test paradigm with multiple sources of cognitive effort.

However, the steeper decrease of PPD in repeat-with-recall condition compared to repeat-only condition was unexpected. PPD has been shown to be sensitive to memory load, therefore, with more words to be remembered, we expected PPD to increase accordingly over time [16, 27, 28]. Decrease in PPD was reported when listeners tended to give up in the tasks that were impossibly difficult [29, 31]. In those cases, performance level was typically low (around 0%). But we did not observe a decrease in recognition and recall performance for words in the later part of the list in our results, or a worse word recognition performance in repeat-with-recall condition at difficult 0dB condition (in fact, word recognition was higher in repeat-

with-recall than repeat-only condition). This suggests that listeners did not give up at the later part of the word list, or at 0dB. Similarly, a smaller PPD at 0dB in repeat-with-recall than repeat-only condition was surprising. Additional recall task with difficult SNR is certainly more demanding than a single task, therefore, we expected PPD to be larger in the repeat-with-recall condition and at difficult SNR level. But we observed the opposite: PPD actually decreased in the repeat-with-recall condition. We do not believe that these are spurious results. This huge contrast with the well-established effect of task demands on the pupillary response was also observed in Zekveld et al. [45]. In Zekveld et al. [45], participants had to recall the four-word cues (either related or unrelated to the following sentence) presented visually before the onset of the sentence embedded in speech masker. The 7dB SNR difference between two sentence-in-noise conditions (-17dB and -10dB) elicited a difference in intelligibility, but not in peak and mean pupil dilation. Zekveld et al. [45] interpreted the absence of pupillary difference between two SNRs as participants prioritising the central factors (memory task) than peripheral factors (sentence recognition task). There are a few characteristics that distinguish our design from Zekveld et al. [45]. Firstly, the memory and sentence recognition tasks in Zekveld et al. [45] were more independent: participants read the cue words for 5s before the auditory stimulus onset; after the auditory stimulus offset, participants either repeated the sentence or the cue words. This separation between two tasks could facilitate intentional prioritisation of the memory over the speech recognition task. Secondly, participants in Zekveld et al. [45] only needed to memorise a four-word cue at the start of each trial, with no accumulation of memory load over time. In comparison, the memory task in our paradigm was more imposing on the limited cognitive resources: participants had to complete both word recognition and memorising tasks within the same time window, and they needed to keep retaining more words in the memory from the 1st to the 10th word. Therefore, it is not surprising that we observed not only a lack of correlation between task demands and pupillary response at easier SNR levels, but also a reversal of that relation at the most cognitively demanding condition (0dB and repeat-with-recall).

One explanation for the steep decrease of PPD in sustained listening condition could be due to fatigue. In a similar sustained listening condition, McGarrigle et al. [44] asked NH participants to listen to two short passages of text with multi-talker babble noise at either -8 dB and 15 dB, and at the end of each passage judge whether images presented on the screen were mentioned in the previous passage. A steeper decrease in (normalised and baseline corrected) pupil size during listening was found for difficult SNR than easy SNR, but only in the second half of the trial block. This was interpreted as fatigue kicking in at the second section of the test. It is likely that in our study, the steeper decrease of PPD in repeat-with-recall condition could also be the sign of overload and fatigue with continuing effort to recognise, encode and rehearse isolated words. However, the decreasing trend reported in McGarrigle et al. [44] was not found in McGarrigle et al. [79] when using a similar test for school-aged children, so it is still unclear how reliably and accurately this metric is related to fatigue.

A more likely explanation to the steeper decrease of PPD in repeat-with-recall condition is that the dynamic range of the pupil could be constrained by baseline diameter. Critically, for the first word in the list, PPD was bigger in repeat-with-recall than repeat-only condition but the baseline diameter was similar. As the baseline diameter grew bigger and plateaued in repeat-with-recall condition, PPD did not have much space to grow, so it decreased faster than repeat-only condition. Similarly, at repeat-with-recall condition, baseline diameter was already bigger than the repeat-only condition for all SNR levels to start with, leaving little room for PPD to increase further during the task. It looks as if under sustained listening condition, there is a limit on the magnitude of pupil dilation, beyond which no further increase is possible. This limit must not be imposed by physiological constraint of the iris muscles, because at

the onset of the recall, pupil diameter increased dramatically, on average by 0.3mm or equivalent to an effect six times bigger than the average PPD at the 10th word (also seen in Cabestrero et al. [28] and discussed in Zekveld et al. [45]). Instead, this limit might be of a cognitive origin rather than the physiological constraint. Similarly, Puma et al. [80] reported a similar ceiling in EEG alpha and theta band power when participants were overloaded with multiple concurrent tasks. This limit might be associated with the saturation in cognitive resources allocation. In order to ensure successful retrieval of words from long- and short-term memory storage at the recall stage, some cognitive resources should be preserved and held until the later part of the test. Therefore, as memory load accumulated (increase in baseline diameter) and approached the limit allocated for the recognition and encoding stage, fewer new resources would be assigned (decrease in PPD), so that enough resources were reserved for the recall stage. The reserved cognitive resources were finally put to use at the onset of recall, leading to a big 'jump' in pupil diameter.

This could be a phenomenal illustration of how cognitive resources are managed in a highly flexible and goal-directed manner. More importantly, as demonstrated in our experiment, this cognitive planning is reflected in the pupillary response. When listeners need to reserve some cognitive resources for later tasks, the pupillary response might show a cap until the next task. In Cabestrero et al. [28], the biggest 'jump' at the onset of recall was when 5 digits were to be recalled (low load), and the smallest 'jump' was when 11 digits were to be recalled (overload), suggesting that this sharp increase in pupil diameter is proportionate to the cognitive resources left for the recall task. Arguably, how cognitive resources are allocated to different tasks could also depend on individual cognitive capacity and cognitive abilities. Listeners with bigger cognitive capacity and better abilities to process speech in noise, might allocate fewer resources (lower limit) to word recognition and encoding, because they will be more efficient in completing the task [59, 81]. In this case, listeners might show a bigger increase in pupil diameter at the onset of recall because they have more cognitive resources left for the recall section. To fully test this hypothesis, future studies need to include more individual cognitive ability measurements and different types of manipulations on cognitive load (for instance, manipulating the memory load by varying the number of words to recall).

## 3.3 Pupillary response to word recognition and memory

Baseline pupil diameter reflected the accumulation of memory load from one serial position to the next. On an individual level, baseline diameter was also related to recall performance, as shown by their significant correlation.

Bigger PPD and more delayed dilation for incorrectly than correctly repeated words in repeat-only condition is also observed in other studies using sentence stimuli [17, 21, 29]. But in the condition requiring heavy and sustained effort (repeat-with-recall), PPD saturated too quickly, especially later in the word list, to support the correlation with word recognition. It seemed that the dynamic range of pupillary response was constrained by the baseline diameter. This further highlights the issue aforementioned, namely that the saturation in pupillary response under sustained load might make PPD problematic for quantifying the actual effort.

Nevertheless, PPD remains a reliable index of listening effort during a single listening task and a potential biomarker for memory processing. Typically, when comparing the recall performance, we found words that were successfully recalled had bigger pupillary response than those forgotten at the encoding stage during time window 2. Papesh et al. [82] suggested a similar relation between PPD and memory encoding success: words that were remembered with higher degree of confidence showed bigger PPD, relative to words that were remembered with less confidence or forgotten. Kucewicz et al. [54] showed that subsequently

recalled words had higher peak pupil dilation 1s after the onset of words being presented visually on the screen for memorising. This is at a similar time point as the difference we observed in the time window 2.

Taken as a whole, these results picture a complex story of the allocation and dynamics of cognitive resources during speech perception and memory task. Failure to recognise the word is associated with more effortful processing, possibly because more lexical competitors are activated for explicit decision when listeners fail to decode the acoustic signals without ambiguity. This might also initiate retroactive corrective processing that would keep the effort elevated post-stimulus [22]. When words need to be remembered for the recall task, the memory encoding probably becomes a priority after completing the word recognition. If more cognitive resources are expended at this stage to encode the word in the working memory storage, there is a higher chance that it will be retrieved successfully later.

## 3.4 Individual differences

Behavioural performance was correlated with pupillary response, but in different manners: better word recognition performance was related with smaller PPD; better stated word recall performance was related with bigger baseline diameter; bigger baseline diameter was related with easier subjective rating; better word recall performance was related with easier subjective rating. Consistent with the results discussed above, these suggest that different metrics of pupillary responses might relate to different cognitive processing. PPD was an indicator of transient effort expended for decoding the words presented in noise, hence correlated with the word recognition performance. Listeners' subjective feeling is affected both by external task demands (SNR levels and TASK), and one's evaluation of recall success. Note that all three measures (pupillary response, word recall performance and subjective rating) also significantly correlated with age, making it possible that the correlations observed were due to a latent variable, for instance individual cognitive capacity [24, 29, 47, 57, 58, 83–85].

To summarise, while behavioural performance (i.e., recall) and subjective rating indicate the final outcome of a series of cognitive processes, pupillometry can reveal the difference in listening effort between conditions, the temporal dynamics of different stages of cognitive processing, as well as the allocation policy of cognitive resources. However, the present findings highlight that there are still many open questions about what the pupil dilation reflects. Only a handful of studies have looked into the dynamics of pupillary response in realistic conditions, where listening is not the only task demanding cognitive resources. The lack of research makes it difficult to develop theories and methods to disentangle the rich information pupillary responses contain. The current experiment is a good example to enrich our knowledge on the topic, by showing the importance of looking at pupillary metrics (time-series variations, baseline diameter) other than PPD when investigating listening effort under sustained memory or other cognitive loads. In a nutshell, here we found that the baseline carries critical information about the overall level of engagement of cognitive resources and the moment-by-moment allocation of these resources in a complex task. As such it might potentially be used as a predictor for the likelihood of success in a memory task on an individual level. In contrast, the PPD might be a good indicator of the success of word decoding and potential biomarker for memory processing, specifically at the level of a single word. But PPD as an indication of listening effort is not as robust in a complex task when PPD would be constrained by the elevated baseline diameter induced by concurrent tasks and when listeners employ different resource allocation strategies. Therefore, future pupillary metrics and analysis pipeline should, similar to our method, devote more attention on the trial-by-trial variation patterns of baseline diameter.

### 3.5 Limitation

Pupil recordings during word repeat and recall were inevitably contaminated by movements during speech production and involuntary eye movement. No algorithm has been developed yet to reliably adjust pupil diameter for these factors. Special care was taken during the experiment and data preprocessing: participants were instructed to keep fixating at the fixation circle during verbal responses; we extrapolated points in the pupil traces where the centre of gazing was beyond 3SD from the centre and excluded trials where over 20% of the traces were either blinks or erratic gazing. Although this lead to loss of data, we ensured that the data left for analysis was valid.

Nevertheless, speech production following the response cue could potentially interfere with the pupillary response corresponding to memory encoding. Individual differences in the timing of responding could also interfere with the correspondence between memory encoding and pupillary response. However, this artefact was present for every word because participants needed to repeat words in all conditions. Therefore, the difference in pupil trace observed within this time window could not be entirely due to production confounds.

### 4 Conclusion

As one of the first few studies to investigate pupillary responses under sustained and complex listening condition, the present study serves as a bridge between established listening effort research and future direction of understanding and quantifying listening effort in real-life communication in various populations. The concurrent recall task did not allow listeners to process just one item, shake off the load once finished and start afresh for the next item. Instead, they needed to be constantly attentive and allocating cognitive resources to process new items while holding other information in (working) memory. This is similar to a real-life communication scenario where multiple tasks compete for a limited pool of cognitive resources over a period of time. Our results suggest that PPD, a traditional pupillometry metric for listening effort in a single listening task, is not robust anymore in a complex task. Instead, the baseline and specifically its trial-by-trial variations are more indicative of the overall cognitive load. Results suggest that both the magnitude and temporal pattern of pupillary response differ greatly in sustained listening condition from those in a single task.

Although real-life speech communication is even more complex and dynamic, the present study serves as a good starting point by choosing a paradigm that could provide enough approximation to cognitive processing in speech communication, yet sufficient time locking to a given type of cognitive processing to ensure the interpretability of the results. A better understanding of listening effort in ecological environments is also important for developing clinical measurement, especially for CI users and HI listeners. It is possible that prior motivational, emotional, cognitive factors and social pressure could disturb the relation between pupillary response and listening effort that is well-established in research settings.

### Supporting information

**S1 Appendix. Alternative method to calculate PPD.** Results and discussions on the alternative method to perform baseline correction using the averaged pupil trace 1s before the first word in the list.
(PDF)

**S2 Appendix. Model summary outputs.** Model parameter estimates and model comparison statistics for the best fitting models. The reference level for the categorical factor LISTENING

is 0dB, for the factor TASK is repeat-only.
(PDF)

**S3 Appendix. Position effect in the word recall task.** Analysis on the position of the words recalled in the repeat-with-recall task.
(PDF)

**S1 Raw data.**
(ZIP)

## Acknowledgments

We thank Florian Malaval and Arthur Delage for assistance with running the experiment with native French-speaking participants.

## Author Contributions

**Conceptualization:** Yue Zhang, Alexandre Lehmann, Mickael Deroche.

**Data curation:** Yue Zhang, Mickael Deroche.

**Formal analysis:** Yue Zhang, Mickael Deroche.

**Funding acquisition:** Alexandre Lehmann, Mickael Deroche.

**Investigation:** Yue Zhang, Alexandre Lehmann, Mickael Deroche.

**Methodology:** Yue Zhang, Alexandre Lehmann, Mickael Deroche.

**Project administration:** Yue Zhang, Alexandre Lehmann, Mickael Deroche.

**Resources:** Alexandre Lehmann, Mickael Deroche.

**Software:** Yue Zhang, Alexandre Lehmann, Mickael Deroche.

**Supervision:** Alexandre Lehmann, Mickael Deroche.

**Validation:** Alexandre Lehmann, Mickael Deroche.

**Visualization:** Yue Zhang.

**Writing – original draft:** Yue Zhang.

**Writing – review & editing:** Alexandre Lehmann, Mickael Deroche.

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
