## [Decision Letter · Decision Letter 0]

15 Jun 2020

PONE-D-20-12674

Disentangling listening effort and memory load beyond behavioural evidence

PLOS ONE

Dear Dr. Zhang,

Thank you for submitting your manuscript to PLOS ONE. After careful consideration, we feel that it has merit but does not fully meet PLOS ONE’s publication criteria as it currently stands. Therefore, we invite you to submit a revised version of the manuscript that addresses the points raised during the review process.

We look forward to receiving your revised manuscript.

Kind regards,

Claude Alain

Academic Editor

PLOS ONE

Journal Requirements:

2. Please amend either the title on the online submission form (via Edit Submission) or the title in the manuscript so that they are identical.

3. Please provide additional details regarding participant consent. In the Methods section, please ensure that you have specified (1) whether consent was informed and (2) what type you obtained (for instance, written or verbal). If your study included minors, state whether you obtained consent from parents or guardians. If the need for consent was waived by the ethics committee, please include this information.

4. Please ensure that you have outlined how you recorded data on pupil diameter in your Methods section.

Additional Editor Comments (if provided):

Reviewers' comments:

Reviewer's Responses to Questions

**Comments to the Author**

1. Is the manuscript technically sound, and do the data support the conclusions?

Reviewer #1: Yes

Reviewer #2: Yes

2. Has the statistical analysis been performed appropriately and rigorously? 

Reviewer #1: Yes

Reviewer #2: Yes

3. Have the authors made all data underlying the findings in their manuscript fully available?

Reviewer #1: Yes

Reviewer #2: No

4. Is the manuscript presented in an intelligible fashion and written in standard English?

Reviewer #1: Yes

Reviewer #2: Yes

5. Review Comments to the Author

Reviewer #1: Review on Zhang et al. “Disentangling listening effort and memory load beyond behavioural evidence”

In this work, the authors aim to characterize the effect of listening effort and memory load on pupil diameter using a dual-task paradigm. They reported the behavioural and pupillometry result from one experiment, where subjects were instructed to repeat a word immediately after its representation. Listening effort was manipulated by varying the SNR of the word. Memory load was manipulated by having an additional task where subjects also needed to memorise 10 words in a row and recall these 10 words after the 10th word. After the response, subjects self-reported their subjective rating of effort.

Most of their findings were consistent with previous studies or very much expected: (1) the memory recall performance decreased in the more difficult listening condition (i.e. the word is hard to hear), (2) larger pupil dilation associated with harder-to-hear words due to the greater listening effort and associated with words in the dual-task due to the greater cognitive load, (3) larger pupil dilation should be associated with response (verbally produce the word), especially in the more difficult listening condition. (4) higher subjective effort rating in harder conditions. The authors were surprised by the fact that although the absolute pupil diameter was larger in the dual-task over the 10-word trial, one of the pupil metrics, peak pupil diameter (=max pupil diameter – pre-trial baseline), decreased since the second word. From this, they claim that the involvement of extra cognitive load interferes with the effect of listening effort on pupil diameter and this is possibly due to the fatigue.

Overall, I found the study well-designed and the data carefully analysed. I particularly like the dual-task paradigm used here, which is quite elegant; the two conditions both involve same acoustic stimuli (i.e. word) and basic task (i.e. recognize and repeat the word) but differ only in the need of maintaining the memory of words. This makes this study distinct from the past studies (as authors mentioned around line 514). This makes the comparison of pupil data neat and clear. The paradigm is also of great potential as it is very close to the real-life listening situation: listeners to recognize the word in noise, reproduce the word accurately, and occasionally remember the word for future recall.

***Major concerns***

1. The possibly most important finding of this study is the fact the PPD (peak pupil diameter) tends to be smaller in the last few words in the repeat-to-recall condition. The authors were very surprised by this result and tried to interpret it by comparing with the previous listening effort studies like Zekveld et al, 2019. However, the explanation the authors offered in the discussion was extensive but not satisfying. It has been well-known that PPD is not only related to the effort or load but also strongly related to its baseline; the larger the baseline, the smaller the PPD. Figure 4b clearly showed that the large baseline is the case. Thus, a simple explanation for this result is that pupil simply saturated in the repeat with recall condition and the pupil simply cannot expand further in the presence of additional words and responses. If this is the case, the result is not surprising at all.

To exclude this possibility, the authors should consider running further analysis (e.g. regress out the effect of baseline from PPD) or conducting additional experiments to show that pupil still CAN dilate further in the repeat with recall condition. If these cannot be done, the authors should at least discuss it in the discussion. The saturation could be not only due to the mechanical limitation of the muscles controlling the pupil diameter, but also because pupil diameter is strongly correlated with the norepinephrine activity in the LC system. Since the authors are aware of the link between pupil diameter and LC-NE system as this was briefly mentioned in Introduction (line 36), they should also take this into account in the discussion.

2. As mentioned before, this paradigm is very neat and of great potential. The authors have already manipulated the level of listening effort using different levels of SNRs. However, the paradigm lacks the manipulation of cognitive load/memory load while it can be simply done. One way to manipulate the memory load is to vary the number of the words required to recall, for example, 5/10/15 words to recall. This will add an additional but necessary dimension to the existing study, otherwise it’s not able to disentangle the effect of listening effort on pupil diameter and that of memory load on pupil diameter, which is actually stated in the title of this paper. This additional experiment with varying memory load should also provide some answers to the questions stated in (1) whether the pupil is saturated in the repeat with recall condition.

3. [Line 549] A minor concern related to this part is that Zekvel et al., 2019 might not be the best study to compare with. The paradigm used in the current study requires listeners to sustain attention over a much longer period; 10 words in total including the sound presentation, the word reproduction, and the word type-in by the experimenter might take almost 30 seconds. I would recommend the authors to take a look at recent pupillometry publications using a sound signal with a similar length.

4. [line 122] The authors need to justify how this sample size was determined.

5. [line 240] How was PPD computed here? Was it extracted from each trial and then averaged within each subject? Or was PPD directly extracted from each subject’s average pupil diameter response?

6. [line 248] Similar question (5) applies to peak latency.

7. [Line 387] The stat test shows a significant difference in the second time window. However, by looking at Figure 5b the error bar of Forgotten and Correct largely overlaps and makes this test result unconvincing. Could the authors run a time-series stat analysis on the pupil data (like the analysis used in Figure 3a, Zhao et al. 2019 Trends in Hearing) to double-check whether the significance is true and if so when the significant interval starts?

8. [line 410] Please state the method of the correlation. e.g. Spearman or Pearson?

***Other comments***

1. [Fig.5] Please use different colours for different conditions’ shaded area. The current colour and pattern makes it hard to tell which area belongs to which condition.

2. [Fig.5] How are these time windows determined?

3. [Figure 6] The solid curve looks over-smoothed compared to its shaded area. Comparing the relationship between the shaded area and the solid curve in Figure 5, the shaded area in Figure 6 are extremely spiky. Maybe additional smooth was accidentally applied to the group average? If so, the authors should justify the difference in the analysis pipeline.

4. [line 394] Possibly I misunderstood the content, but could the authors please provide more details or make it clearer: how is the mean pupil diameter computed here? Over which time window? Also, to support the statement this line, could you also plot the mean pupil diameter against the number stated words and test the correlation like Figure 7?

5. [line 417] Nice to see that the authors noticed the relationship between these metrics with the age as ageing is a known factor in pupil diameter. As the authors stated “Note that these correlations should be considered with caution due to no corrections” I was expecting to see these results with the age being regressed out.

6. [line 472] “the recall task probably because it was more interesting and rewarding”. It’s unclear how the recall task can be rewarding here. Did the authors apply a special bonus to the repeat with recall condition?

Reviewer #2: I think this is a pretty good paper, in principle. The work seems to have been well done, and it addresses some very interesting and timely questions. That said, I found the original manuscript relatively difficult to read, not because of any problems with language but because I think it needs one more thorough revision now that the authors have successfully thought through all of their ideas - there are good ideas in here, but they're jumbled up still, and hard to find. Thus, most of my suggestions relate to writing and organization.

Abstract

When stating “different signal to noise ratios” please state what these are. Otherwise it makes no sense to state that “ (PPD) was bigger in the 0dB versus other conditions” because we have no idea whether those were all negative, or all positive, or by how much compared to 0 dB.

It’s not clear to me from the abstract how baseline PPD can follow a growth pattern. I assume this means across trials or perhaps across words within a trial, but that is not clear.

If PPD increased, how did it them decrease? This whole paragraph could be rewritten more clearly.

No need to refer to “concur with the recent literature” in the abstract. IT's a statement that is too vague to be useful.

11 speech recognition is similar to what?

48-52 weird change of tense in the middle of this sentence. (“needs to decode… pondering…”)

93+ I think the section on LISTENING condition should be a separate paragraph. Also, I’m not sure it makes sense to go into this much detail about the study before the methods section. Right now, I’m left wanting to know more – for example, was the SNR variation blocked or was it randomized within trials? So either more info is needed here, or possibly less.

95-97 this sentence seems out of place (“The effect of SNR …”)

98-99 the issue of pupil traces for recalled vs. not recalled tokens seems quite a bit different from the topics that have been discussed so far. This is a big issue in the memory literature, and you probably need to take a look as some of that to restructure your introduction to better reflect this emphasis and the scientific context in which it fits.

105 I’d leave out “according to past studies” – you’re doing your own work here.

109-113 This prediction needs to be broken up, and perhaps re-thought. If baseline PD is expected to increase with memory load then it’s not obvious to me that PPD will also increase – if you’re raising the floor over the course of the 10 items in the trial, does it make sense to assume that the peak from that increasingly higher floor will *also* be increasing? Also, How do you think the increased memory load-related increasing baseline will interact with the previously mentioned decrease in PD as a function of time-on-task?

114-116 It is not clear whether this “rise” in PD is referring to the appearance of the pupil diameter trace in the course of a single stimulation, or over the course of a 10-word trial, or over the course of the entire experiment. The issue of time needs to be MUCH clearer throughout the paper.

121-126 It’s common to provide a gender breakdown for participants. But, more importantly, it’s incredibly important to identify the number of participants run in French and the number run in English. This could be a very significant factor and should probably be included in the final analysis [it is analyzed, so definitely mention it here].

128-138 How many words in all? How many lists per condition? This is starting to seem like a rather incomplete Stimuli section.

127-134 Please break down durations by language. Ideally, please list all words in the supplementary materials. Was there any attempt to balance the intelligibility of the different word lists? If not, word list might need to be a factor in the eventual model as well. Also, a reference to “Fournier” words is needed, I don't know what those.

147 Was the calibration done with a pure tone, or with the speech stimuli and (if speech) then with or without noise (and if with noise, then at what SNR level)?

150 I presume “at 14 dB” means “at 14 dB SNR” but that should be made clear

159 what is the significance of the (0.5s) in this line and the (1s) in the next line?

From the procedures section, it’s still not clear whether SNR was constant for a given set of words, or block of trials, or randomized (either across words within a trial, which would be admittedly strange) or across trials. Also, the ordering of TASK and LISTENING condition combinations is not specified.

162 Adjustment of the Target level in a constant noise level means that in the highest SNR condition the Target was 14 dB louder than in the 0 dB condition, right? This seems like an extreme difference, even without the presence of noise. What was the noise level alone? More importantly, what was the level of the signal in the QUIET condition – the same as that in the 0dB condition, or the 14 dB condition, or something else? Given that autonomic responses can be influenced by absolute level, what procedures were implemented to ensure that the differences observed were not simply due to differences in overall signal level?

169 Were there procedures for dealing with homophones? Were the transcribers well-versed in the set of words being used? Could the experimenter/transcriber see the intended word?

175 what was the time delay before the word RECALL appeared?

184 Was a “block” one set of 10 words, or 3?

189-192 Please provide degrees of freedom for the t-tests.

203 missing “were” before “retained” (or change to “remained”)

1.4.1+ It would be easier to understand the statistical analyses if you would provide the actual model, either in lme4 syntax (easy to do in this case) or in standard mathematical notation. This is quite common nowadays and could be put into supplementary materials if space is an issue.

232 What does “aggregated per word” mean? Averaged?

Also, in this context, it is confusing to say “per word” if you actually mean (as I think you do) “per word position” (i.e. 1-10). Aggregating “per word” seems impossible if listeners never heard the same word twice as implied in the methods section.

236 Please clarify – these were the “aggregated” traces, right? 1 trace per subject per word-position?

Also, given that the actual words were presumably of different durations, I think using absolute time (i.e. in seconds) is a bad idea, because it could blur effects that are related to the duration of the word. You might consider normalizing all times before any averaging is done. Or fit a curve to each individual trace and then compute the peak and the latency from that, then do averages over those values.

243-26 I like the comparison between the “block baseline” and the “word position baseline”.

252-291 I find it a little confusing (and quite demanding on my working memory) to present all statistical analyses prior to any results. I think this section would be helped by using sub-headings and, again, by providing the actual models in either lme4 syntax or as an equation. Alternatively, these paragraphs could be put as the initial paragraph of the respective results (sub)sections.

265-282 I think the discussion of the second time window suggests that what you really should be doing is looking at the entire pupil diameter curve from the onset of the word-position to 1.5s after its offset. See Winn & Moore (2018) for a really clever way of breaking such long(ish) traces down for analysis.

Figure 2 a minor point, but it seems needlessly complicated to present the results with different Y axes representing essentially the same thing - % correct words repeated vs. Average number of words remembered (presumably out of 10?)

323-418 The results section is incredibly hard to read. Please revise to put things into complete sentences. It’s not just about presenting a bunch of equations here, you need to organize them in such a way that the reader can understand what you’re talking about. At this point I can’t really. Please give values (i.e. don’t just tell me baseline PD was bigger in one level than another, tell me what the value was for each level). It’s confusing to read that baseline pupil diameter was “bigger than 14 dB” (line 332) when to my knowledge we don’t typically measure pupil diameter in decibels. Yes, I can figure out what you mean, but this is currently written as it might be written in a lab notebook, for personal consumption, not as it should be written for scientific communication. And in some cases, it’s opaque: in line 335, is the (0.2 mm) referring to the absolute diameter, or the amount by which it is bigger? Even at the end “due to no corrections” is practically txting the results…

Also, I think the trend analyses could be discussed separately. Basically, right now it seems as if you're more or less just listing the results of all the tests you did, maybe in chronological order or perhaps loosely organized (?) according to dependent measure. Please consider some way of organizing the results in a way that facilitates the reader’s understanding of why you conclude what you will eventually conclude or, at a minimum, that reflects the issues that you determined were relevant to investigate as described in the introduction. Ideally, the results section should be presented in the same order as the discussion section, which should walk the reader through the data toward the eventual theoretical claims that you want to make (and which should reflect the relative importance of topics as discussed in the introduction). Right now I honestly can’t figure out what data point(s) are particularly relevant or irrelevant, it just sort of devolved into a giant mass of statistical tests presented without obvious organization.

[Discussion section is also a bit confusing - mostly due to digressions, though]

Figures 3 & 4 Looking at the traces in Figures 3a and 4a it seems apparent to me that peak pupil dilation may not be a useful metric here. Except in the first word position there really isn’t much of a *peak* of any sort visible in 4a. And you can see that when those word positions get averaged together (for the images in 3a) any potential peakedness disappears. So why not use average PD or something like that? I think that would tell the story at least as well, and would be less subject to potentially weird micro-effects such as the weird flip of the black and red dots in positions 5, 8, and 9 of figure 4c.

Also, the Y axis of 3c and 4c should somehow indicate that this is change from baseline.

In general, I’d recommend considering a very different way of doing this analysis, perhaps along the lines of Winn & Moore (2018).

340 should this be 3b or 4b?

432-443 references needed here to Pichora-Fuller et al. 1995, Surprenant (1999, 2007).

461-506 I really struggled with this discussion. I think the long and detailed references to the noise reduction work are distracting and superfluous. So lines 436-455 could be reduced to just lines 451-455.

Also, this discussion brings up the question of what, exactly, pupil dilation tells us. Arguably, it could provide information about the overall level of engagement of cognitive resources (I think that’s what the baseline measurement is supposed to get at, here) in these two conditions, as well as the moment-by-moment allocation of those resources during part of a task (encoding, repetition, recall). Given that you *have* pupil dilation data, I think this needs to be addressed somehow, before going into details of what dual task paradigms may or may not tell us.

And, finally, what do you conclude? I appreciate that there are multiple possible interpretations, but you've thought about this far more than most. Could you lead the reader from this apparent bafflement into something that we can be more satisfied with?

488 Could you examine age differences in your data? What would you predict to see either in terms of behavior or pupil dilation if people are prioritizing things differently?

490 what does it mean that the recall paradigm is from previous studies? Which recall paradigm?

498 Define SWIR acronym.

509 What second hypothesis? There are so many hypotheses swirling around by now I’ve lost track of which one is which. Please restate.

519 these references did not all use the same speech perception task. Clarify.

523 You don’t really have data showing any greater effort of your task over other tasks.

540 The lack of position effect is extremely unusual for a serial recall task and needs to be discussed in much more detail. It should also be presented in the results. This is one of those memory effects that is so basic it’s taught in intro psych textbooks... I would very much like to see a graph of word recognition and recall by word position. I have great difficulty imaging that there wasn’t some kind of recency effect at least, if not also a primacy effect, with a 10-item list to be recalled.

585-621 It seems to me that the best explanation for smaller growth of the PPD is that the baseline is increasing. So the limit (probably physiological, based on light levels) is imposed not in terms of how much the pupil can dilate, but in terms of how much of a dilation it will reach. In other words, illumination, which you held constant, may have imposed an upper limit on pupil dilation, such that as the baseline creeps up with increasing memory load in the recall condition, or creeps down with increasing habituation in the repeat-only condition, you get the difference between the two gradually shrinking (in the recall condition) or increasing (in the repeat only condition).

623-626 Word choice seems problematic. What does it mean to “hold predictive power” or to be “responsive for recall performance”? Say what you want to say in a simple way.

References that should be incorporated into a revision

Goldinger, S. D., & Papesh, M. H. (2012). Pupil dilation reflects the creation and retrieval of memories. Current directions in psychological science, 21(2), 90-95.

Kucewicz, M. T., Dolezal, J., Kremen, V., Berry, B. M., Miller, L. R., Magee, A. L., ... & Worrell, G. A. (2018). Pupil size reflects successful encoding and recall of memory in humans. Scientific reports, 8(1), 1-7.

Miller, A. L., Gross, M. P., & Unsworth, N. (2019). Individual differences in working memory capacity and long-term memory: The influence of intensity of attention to items at encoding as measured by pupil dilation. Journal of Memory and Language, 104, 25-42.

Pichora‐Fuller, M. K., Schneider, B. A., & Daneman, M. (1995). How young and old adults listen to and remember speech in noise. The Journal of the Acoustical Society of America, 97(1), 593-608.

Surprenant, A. M. (1999). The effect of noise on memory for spoken syllables. International Journal of Psychology, 34(5-6), 328-333.

Surprenant, A. M. (2007). Effects of noise on identification and serial recall of nonsense syllables in older and younger adults. Aging, Neuropsychology, and Cognition, 14(2), 126-143.

Winn, M. B., & Moore, A. N. (2018). Pupillometry reveals that context benefit in speech perception can be disrupted by later-occurring sounds, especially in listeners with cochlear implants. Trends in hearing, 22, 2331216518808962.

6. PLOS authors have the option to publish the peer review history of their article (what does this mean?). If published, this will include your full peer review and any attached files.

Reviewer #1: Yes: Sijia Zhao

Reviewer #2: Yes: Alexander L. Francis

---

## [Author Response · Author response to Decision Letter 0]

21 Aug 2020

All line numbers in our response are referring to the marked-up manuscript for clear comparison with the original draft.

We thank the editor and reviewers for their comments. We have incorporated their feedback in our revised article to improve the methods reporting, organise better the sections, update with more relevant literature and clarify our main findings.

In summary, we believe that we have strengthened our research article to meet your publication criteria.

Reviewer #1: Review on Zhang et al. “Disentangling listening effort and memory load beyond behavioural evidence” Overall, I found the study well-designed and the data carefully analysed. I particularly like the dual-task paradigm used here, which is quite elegant; the two conditions both involve same acoustic stimuli (i.e. word) and basic task (i.e. recognize and repeat the word) but differ only in the need of maintaining the memory of words. This makes this study distinct from the past studies (as authors mentioned around line 514). This makes the comparison of pupil data neat and clear. The paradigm is also of great potential as it is very close to the real-life listening situation: listeners to recognize the word in noise, reproduce the word accurately, and occasionally remember the word for future recall.

We thank reviewer 1 for the very useful feedback on the manuscript. We are addressing each concern in the following:

***Major concerns***

1. The possibly most important finding of this study is the fact the PPD (peak pupil diameter) tends to be smaller in the last few words in the repeat-to-recall condition. The authors were very surprised by this result and tried to interpret it by comparing with the previous listening effort studies like Zekveld et al, 2019. However, the explanation the authors offered in the discussion was extensive but not satisfying. It has been well-known that PPD is not only related to the effort or load but also strongly related to its baseline; the larger the baseline, the smaller the PPD. Figure 4b clearly showed that the large baseline is the case. Thus, a simple explanation for this result is that pupil simply saturated in the repeat with recall condition and the pupil simply cannot expand further in the presence of additional words and responses. If this is the case, the result is not surprising at all.

To exclude this possibility, the authors should consider running further analysis (e.g. regress out the effect of baseline from PPD) or conducting additional experiments to show that pupil still CAN dilate further in the repeat with recall condition. If these cannot be done, the authors should at least discuss it in the discussion. The saturation could be not only due to the mechanical limitation of the muscles controlling the pupil diameter, but also because pupil diameter is strongly correlated with the norepinephrine activity in the LC system. Since the authors are aware of the link between pupil diameter and LC-NE system as this was briefly mentioned in Introduction (line 36), they should also take this into account in the discussion.

We thank the reviewer’s contribution to this interpretation of the results.

We agree that baseline and PPD are correlated mathematically because (X-Y) and Y will always be correlated by R2=0.5, assuming X and Y completely random. And past literatures have also demonstrated this correlation. However, the exact relation between baseline and PPD during a hearing or cognitive task depends on the underlying cause. For instance, the effect of old age induces smaller baseline and smaller PPD due to physiological constraints and changes of activity in peripheral and/or central nervous system (Piquado et al., 2010; Kuchinsky et al., 2016; Wang et al., 2018). Lower luminance induces bigger baseline but smaller PPD due to the ‘gripping’ of parasympathetic system (Wang et al., 2018). Therefore, it is unclear what direction this relation between PPD and baseline should be in a task with concurrent listening and cognitive demands. What is surprising in our results is that we initially hypothesised that PPD might increase with more difficult SNR and more items to retain in the memory, but we see instead that the pupil dilation ‘capped’ during the listening section (before the word recall section). However, looking at the baseline dynamics let us understand partially the cause of the ‘capping’. This highlights the importance to look both at the PPD and baseline in future experiments that involves more ecologically realistic tests.

We share with Reviewer 1 the desire to further disentangle baseline from PPD by either regressing out the effect

of baseline or showing that pupil can still dilate further in repeat with recall condition. The first approach, however, is problematic as long as we do not understand the exact conditions where base and PPD are negatively correlated from conditions where they may be positively correlated (whether this is seen within or across subjects). So, we opted for the second approach: while a sort of pupil saturation was present during the listening and encoding section from 1st to 10th word, the pupil increased at the onset of recall on average by 0.3mm! Reviewer 1 did not realize this finding, so we made it more explicit in the article: the ceiling of the pupil during the recall blocks cannot be due to mechanical limitation of the muscles controlling the pupil diameter, because right at the end of the block, the pupil diameter rose considerably, an effect equivalent to six times the average PPD at the 10th word. Therefore, it is clear that the pupil ceiling during listening and encoding was not at all due to mechanical constraints but originated from cognitive resource allocation strategy. The best interpretation we can offer – and that we discussed – is that listeners would reserve their resources during the 1st to 10th word in order to retrieve the words during the recall section.

2. As mentioned before, this paradigm is very neat and of great potential. The authors have already manipulated the level of listening effort using different levels of SNRs. However, the paradigm lacks the manipulation of cognitive load/memory load while it can be simply done. One way to manipulate the memory load is to vary the number of the words required to recall, for example, 5/10/15 words to recall. This will add an additional but necessary dimension to the existing study, otherwise it’s not able to disentangle the effect of listening effort on pupil diameter and that of memory load on pupil diameter, which is actually stated in the title of this paper. This additional experiment with varying memory load should also provide some answers to the questions stated in (1) whether the pupil is saturated in the repeat with recall condition.

We agree with the reviewer that a manipulation on the memory load would in principle add further support to the major point raised in the current experiment. But this is not a trivial thing to do: changing the size of the list will likely not do the trick; it has been done several times in the literature - i.e. different flavors of the SWIR paradigms with varying list sizes – and the problem is that listeners tend to “normalize” their performance. They would perform surprisingly poorly with short lists and surprisingly well with long lists, such that the manipulation supposed to vary the difficulty level within the same task is ineffective. So, this is a good idea in theory but there are hurdles to overcome beforehand that originate directly from the non-linearity, and possibly non-monotonicity, of the response. This is why we took a first step in this study, looking at how drastically different the pupil dynamics are when memory is involved. And we showed that a SNR manipulation – which is not easily compensated by flexible resource allocation – had relatively little impact in the recall conditions. It is likely that the same will hold whether the lists are 5, 10, or 15 words long. This being said, it would be a topic worthy of future investigation, so we added this point to highlight the important next step following up this experiment (line 698).

3. [Line 549] A minor concern related to this part is that Zekvel et al., 2019 might not be the best study to compare with. The paradigm used in the current study requires listeners to sustain attention over a much longer period; 10 words in total including the sound presentation, the word reproduction, and the word type-in by the experimenter might take almost 30 seconds. I would recommend the authors to take a look at recent pupillometry publications using a sound signal with a similar length.

We thank the reviewers’ advice on choosing other publications with similar length stimuli. We have added a few papers in the introduction and discussion section using similar length stimuli (Zhao et al., 2019; Goldinger and Papesh, 2012; Kucewicz et al., 2018 ).

4. [line 122] The authors need to justify how this sample size was determined.

A priori analysis using G*Power3 showed N=26 (three predictors: SNR, recall condition and word position; alpha error probability = 0.05) for an effect size of 0.8 using F test for linear multiple regression.

(Faul, F., Erdfelder, E., Lang, A. G., & Buchner, A. (2007). G* Power 3: A flexible statistical power analysis program for the social, behavioral, and biomedical sciences. Behavior research methods, 39(2), 175-191.)

5. [line 240] How was PPD computed here? Was it extracted from each trial and then averaged within each subject? Or was PPD directly extracted from each subject’s average pupil diameter response?

We thank Reviewer 1 for this comment, which we have repeatedly heard while presenting these results at conferences. It is a matter of constant debate. For sentences, the pupil dilation is more stable than it is for individual words, and thus arguably, one might want to extract PPD directly from individual sentences. We found that this did not apply well to individual words. So, we opted for the PPD taken from the averaged traces. We

firstly performed the baseline correction to subtract the baseline of each trial from the pupil trace. Then traces were aligned by the onset of the response prompt and aggregated per listener per condition. PPD was then calculated at this aggregated level, instead of the trial level. This method was chosen in aligned with past studies and ensured PPD was more robust (Zekveld et al., 2010; Zekveld et al., 2013; Zekveld et al., 2014).

We have re-organised the method and result sections to clarify the detailed procedure (line280).

6. [line 248] Similar question (5) applies to peak latency.

The latency was also calculated on the averaged traces, not on each trial. We have also re-organised the method and result section to clarify the procedure.

7. [Line 387] The stat test shows a significant difference in the second time window. However, by looking at Figure 5b the error bar of Forgotten and Correct largely overlaps and makes this test result unconvincing. Could the authors run a time-series stat analysis on the pupil data (like the analysis used in Figure 3a, Zhao et al. 2019 Trends in Hearing) to double-check whether the significance is true and if so when the significant interval starts?

We appreciate Reviewer 1’s concern on the overlapping error bars in Figure 5b. This was mostly due to plotting pupil traces using raw pupil diameter in mm. Traces were not baseline-corrected by each trial here, so plotting the average of all individuals did not illustrate within-individual differences effectively. We replaced Fig.5b with a baseline-corrected comparison. Raw traces have their merit though, so we still keep the original Figure 5a in mm

to fully illustrate the baseline and trace separation between repeat-only and repeat-with-recall condition.

We thank the reviewer for pointing us to the time-series analysis method in Zhao et al., 2019. Such analyses can delineate accurately the point of separation between two signals, provided there is a tight control of timing for the successive events. As pointed out in the limitation section, we didn't ask participants to respond as quickly as possible, in purpose, because applying a time pressure for the memory task would further affect pupillary response. So the pupil trace after the onset of response contains individual differences in speech production timing, speech production confounds and memory processing. Therefore, the onset of the trace separation seems to us too convoluted by cognitive events. Instead, we used the time window of 1.5s, based on the successful memory encoding effect observed in Kucewicz et al., 2018 with similar stimuli length and timing, which seems more relevant here.

Nevertheless, we would intend to employ more sensitive time-series analysis method as in Zhao et al., 2019 in future studies with better timing control. Specifically, in a follow-up experiment (yet unpublished), we asked participants to NOT repeat but recall, so that we had a 'clean' trace without the effect of speech production and its variable timing. We will then be able to tackle more seriously the onset of separation. Once again, our current view was to demonstrate, first at a qualitative level, that a bigger PPD is not always a bad sign. Many readers would find this result highly controversial because indeed in listening-only situations, a bigger PPD is a reliable sign that

speech was misunderstood (Fig5a).

8. [line 410] Please state the method of the correlation. e.g. Spearman or Pearson?

We used Pearson correlation. Thanks for pointing this missing information out. We have further specified this detail (line431).

***Other comments***

1. [Fig.5] Please use different colours for different conditions’ shaded area. The current colour and pattern makes it hard to tell which area belongs to which condition.

Clarified. We have used both lineshape and colour to differentiate word recall and word repeat performance.

2. [Fig.5] How are these time windows determined?

Time window 1 was determined according to the practice in past studies that looked at the listening effort associated with listening to sound stimuli (line238). Time window 2 was determined by using a similar waiting period (1s) after the event of interest occurred (line273). We hypothesised that time window 2 corresponded to participants rehearsing and encoding the perceived word into working memory storage. Also, Kucewicz et al., 2018 showed that the effect of memory encoding occurred 1s after the visual word presentation, corresponding to the time window 2.

But we noted in the Limitation sections that there are many factors that could interfere with the pupil trace at this time window (speech production, individual differences in the timing of responding/memorisation etc). Therefore, only the comparison between the repeat-with-recall and repeat-without-recall conditions is meaningful because the difference in the event is in rehearsal/memorisation.

3. [Figure 6] The solid curve looks over-smoothed compared to its shaded area. Comparing the relationship between the shaded area and the solid curve in Figure 5, the shaded area in Figure 6 are extremely spiky. Maybe additional smooth was accidentally applied to the group average? If so, the authors should justify the difference in the analysis pipeline.

We thank the reviewer for pointing this missing information out. Indeed, due to fewer traces to average over and great individual differences in recalling behaviour, we had more variable pupil traces (shaded areas), which we wished to reflect for transparency. But, to plot the mean trace (black solid line) we used the default GAM smoothing in ggplot2 package to highlight the general trend of the pupil data.

We have added this detail in the legends of Figure 6.

4. [line 394] Possibly I misunderstood the content, but could the authors please provide more details or make it clearer: how is the mean pupil diameter computed here? Over which time window? Also, to support the statement this line, could you also plot the mean pupil diameter against the number stated words and test the correlation like Figure 7?

We described the procedure to calculate the mean pupil diameter on line333. But we appreciate that the clustered organisation of the method section was unhelpful. We have re-organised the methods and results section so that they are closer. Hopefully this has now eased the comparison.

Upon closer check, we realised that we did not discuss this result later, and due to the limitation mentioned in the Limitation section, we could not interpret too much from this result due to disruptions from the verbal responses. We have now deleted this report to simplify the results.

5. [line 417] Nice to see that the authors noticed the relationship between these metrics with the age as ageing is a known factor in pupil diameter. As the authors stated “Note that these correlations should be considered with caution due to no corrections” I was expecting to see these results with the age being regressed out.

We thank the reviewer for pointing this out. We now add the results of these correlations after regressing out the effect of age (line461).

6. [line 472] “the recall task probably because it was more interesting and rewarding”. It’s unclear how the recall task can be rewarding here. Did the authors apply a special bonus to the repeat with recall condition?

We did not give special bonuses. But for NH listeners, the word recognition task, even at 0 dB SNR, was not very difficult: intelligibility might seem relatively low (at 75% on average) due to the lack of semantic context, but their intelligibility of sentences was 90% at 0 dB (not shown in this article). The real challenge and rewarding section of the experiment was to recall the words because they knew there were always 10 words. Indeed we saw individual differences in how they reacted to their recall performance: some were curious with how many words they recalled correctly and some did not care.

Reviewer #2: I think this is a pretty good paper, in principle. The work seems to have been well done, and it addresses some very interesting and timely questions. That said, I found the original manuscript relatively difficult to read, not because of any problems with language but because I think it needs one more thorough revision now that the authors have successfully thought through all of their ideas - there are good ideas in here, but they're jumbled up still, and hard to find. Thus, most of my suggestions relate to writing and organization.

We thank the reviewer for the detailed and useful feedback on the manuscript. We have addressed the concerns as following:

Abstract

When stating “different signal to noise ratios” please state what these are. Otherwise it makes no sense to state that “ (PPD) was bigger in the 0dB versus other conditions” because we have no idea whether those were all negative, or all positive, or by how much compared to 0 dB.

We have added the specific SNRs.

It’s not clear to me from the abstract how baseline PPD can follow a growth pattern. I assume this means across trials or perhaps across words within a trial, but that is not clear.

We have clarified the growth pattern is within a trial.

If PPD increased, how did it them decrease? This whole paragraph could be rewritten more clearly.

We have used the word ‘variation’ instead of ‘growth’ to avoid ambiguity.

No need to refer to “concur with the recent literature” in the abstract. IT's a statement that is too vague to be useful.

We have deleted that section.

11 speech recognition is similar to what?

We have clarified the sentence (line10).

48-52 weird change of tense in the middle of this sentence. (“needs to decode... pondering...”)

We have corrected the tense here (line50).

93+ I think the section on LISTENING condition should be a separate paragraph. Also, I’m not sure it makes sense

to go into this much detail about the study before the methods section. Right now, I’m left wanting to know more – for example, was the SNR variation blocked or was it randomized within trials? So either more info is needed here, or possibly less.

We have simplified this section(line96).

95-97 this sentence seems out of place (“The effect of SNR ...”)

We have deleted this sentence to simplify this section (line95).

98-99 the issue of pupil traces for recalled vs. not recalled tokens seems quite a bit different from the topics that have been discussed so far. This is a big issue in the memory literature, and you probably need to take a look as some of that to restructure your introduction to better reflect this emphasis and the scientific context in which it fits.

We agree with Reviewer 2 that the literature on the memory is important here, and truly appreciate the suggestions on the relevant memory studies. We have included them in the discussion section (line723), because they will help interpret the pupillary response in the repeat-with-recall condition. We think that the discussion section is more suitable to elaborate on the effect of memory, and reserve the introduction section for highlighting the lack of literature on concurrent cognitive load during listening tasks.

105 I’d leave out “according to past studies” – you’re doing your own work here.

Done.

109-113 This prediction needs to be broken up, and perhaps re-thought. If baseline PD is expected to increase with memory load then it’s not obvious to me that PPD will also increase – if you’re raising the floor over the course of the 10 items in the trial, does it make sense to assume that the peak from that increasingly higher floor will *also* be increasing? Also, How do you think the increased memory load-related increasing baseline will interact with the previously mentioned decrease in PD as a function of time-on-task?

Absolutely, all of these are perfectly valid and important questions, but it has not been easy to articulate these hypotheses before we started this project because of the inherent entanglement between baseline and PPD. This is a point also raised by Reviewer 1 for good reasons: people have a priori assumptions with regard to how PPD should behave if baseline goes in one direction or another. Typically, the idea that PPD will be necessarily restricted if baseline is too high does not hold in many cases. In response to R1, we mention the role of age or luminance as factors that completely disrupt the presumed inverse relationship between the two. Therefore, instead of revising our hypotheses now that we know the results to better differentiate baseline-related hypotheses from PPD-related hypotheses, we think it is more transparent and honest to present the hypotheses as we would have done in a pre-registered format. And what we knew two years ago was that the PPD would increase with adverse SNR, and that the baseline would increase incrementally as listeners rehearse words in their mind within a block, eventually leading to a task difference when averaging over successive trials of a block.

114-116 It is not clear whether this “rise” in PD is referring to the appearance of the pupil diameter trace in the course of a single stimulation, or over the course of a 10-word trial, or over the course of the entire experiment. The issue of time needs to be MUCH clearer throughout the paper.

Sorry for not making this more explicit: we referred to the end of the recall blocks, where listeners were prompted to report to the experimenter all the words they could remember. We clarified this in the hypothesis (line114), and reminded the reader of a similar phrasing on method section 2.6.3.

121-126 It’s common to provide a gender breakdown for participants. But, more importantly, it’s incredibly important to identify the number of participants run in French and the number run in English. This could be a very significant factor and should probably be included in the final analysis [it is analyzed, so definitely mention it here]. We have clarified the gender and language breakdown of the participants (line122).

128-138 How many words in all? How many lists per condition? This is starting to seem like a rather incomplete Stimuli section.

We have clarified these: three different lists per condition and overall 480 words used (line 135 line140).

127-134 Please break down durations by language. Ideally, please list all words in the supplementary materials. Was there any attempt to balance the intelligibility of the different word lists? If not, word list might need to be a factor in the eventual model as well. Also, a reference to “Fournier” words is needed, I don't know what those.

For sure, although the word lists are standardised, there might still be differences. We have now added word list as a random effect in the models, and only kept it in the model if it significantly improved the model fitting using chi- squared test. Details have been added to Supplementary material table 2.

We have added the reference to Fournier corpus. (line132) All words are listed in the references, therefore, we have not listed all words again in the supplementary materials.

147 Was the calibration done with a pure tone, or with the speech stimuli and (if speech) then with or without noise (and if with noise, then at what SNR level)?

The calibration with the headphone was performed using a pure tone 1kHz (line156).

150 I presume “at 14 dB” means “at 14 dB SNR” but that should be made clear

We have clarified this here and at later occurrences.

159 what is the significance of the (0.5s) in this line and the (1s) in the next line?

1s baseline is set following the recommendations in Winn et al., 2018 (between 100ms to 2s), and more specifically for words stimuli in Kuchinsky et al., 2014 Kuchinsky et al., 2013 (1s).

0.5s intertrial is to allow for gradual regress back to the baseline. Although longer intertrial duration is preferable for pupillometry measures, and might be relevant to longer speech material, it is not ideal for the memory task.

From the procedures section, it’s still not clear whether SNR was constant for a given set of words, or block of trials, or randomized (either across words within a trial, which would be admittedly strange) or across trials. Also, the ordering of TASK and LISTENING condition combinations is not specified.

To clarify, what we call a trial is the presentation of a single word. There is no trial with several words. SNR was kept constant for a block of 10 trials. This was necessary to evaluate the effect of SNR when memory was involved. But the rest was fully randomized. We have reorganised and clarified the details in the procedure section (line167).

162 Adjustment of the Target level in a constant noise level means that in the highest SNR condition the Target was 14 dB louder than in the 0 dB condition, right? This seems like an extreme difference, even without the presence of noise. What was the noise level alone? More importantly, what was the level of the signal in the QUIET condition – the same as that in the 0dB condition, or the 14 dB condition, or something else? Given that autonomic responses can be influenced by absolute level, what procedures were implemented to ensure that the differences observed were not simply due to differences in overall signal level?

Yes, target level was at 65 dB at 0 dB, 79 dB at +14 dB SNR, and back to 65 dB in quiet. We have further specified the speech leve at 161.

Fixing the masker noise was chosen following the reference cited in text (Ohlenforst et al., 2018) to avoid participants guessing the upcoming block difficulty. This choice was thus preferable to fixing the target level at 65 dB, especially with a large range of SNRs in the same experiment, similar to the case of Ohlenforst et al., 2018.

In principle absolute level should illicit an automatic impact, but this is not necessarily the case during a listening effort task that can dominate over the automatic response. Also the impact of signal level was not seen in Ohlenforst et al., 2018

169 Were there procedures for dealing with homophones? Were the transcribers well-versed in the set of words being used? Could the experimenter/transcriber see the intended word?

No, remember that the listener did not type the words (we couldn’t do that because the eye gaze had to stay in the middle of the screen). The experimenter could see the correct word in the Matlab interface, so they could type down the correct word instead of its homophones.

We have further specified this in the procedure (line180).

175 what was the time delay before the word RECALL appeared?

We thank the reviewer for pointing this out. We have clarified this delay to be 2s (line187).

184 Was a “block” one set of 10 words, or 3?

Each block contains 10 words. We have clarified this (line195).

189-192 Please provide degrees of freedom for the t-tests.

We have added more details on the t-tests performed (line202) here an other occurances.

203 missing “were” before “retained” (or change to “remained”)

Corrected

1.4.1+ It would be easier to understand the statistical analyses if you would provide the actual model, either in lme4 syntax (easy to do in this case) or in standard mathematical notation. This is quite common nowadays and could be put into supplementary materials if space is an issue.

We have included the best fitting models information in the Supplementary 2.

232 What does “aggregated per word” mean? Averaged?

Yes, we averaged the pupil diameter at the same point in time. This was following the method in past studies and ensured PPD was more robust (Zekveld et al., 2010; Zekveld et al., 2013; Zekveld et al., 2014).

Also, in this context, it is confusing to say “per word” if you actually mean (as I think you do) “per word position” (i.e. 1-10). Aggregating “per word” seems impossible if listeners never heard the same word twice as implied in the methods section.

Thank you for this clarification! We have corrected the wording from ‘word’ to more precise ‘word position’ here and in other occurrences.

236 Please clarify – these were the “aggregated” traces, right? 1 trace per subject per word-position?

Yes, the traces were aggregated/averaged per participant WORD POSITION.

Also, given that the actual words were presumably of different durations, I think using absolute time (i.e. in seconds) is a bad idea, because it could blur effects that are related to the duration of the word. You might consider normalizing all times before any averaging is done. Or fit a curve to each individual trace and then compute the peak and the latency from that, then do averages over those values.

Very nice comment. Indeed, this is a concern for longer speech materials, which typically have bigger duration SD and longer peak latency. In fact, we are doing exactly the same recommended method on sentence stimuli in a manuscript under preparation (sentences at 0, 7, 14 dB and quiet, of different durations).

But this essentially does not apply here to individual words. In the current experiment, the duration SD of monosyllabic word is small (0.09s), and not long enough to disturb the time scale of cognitive driven pupil responses (~ 10Hz).

Perhaps more to the point, the normalisation procedure Reviewer 2 suggested is problematic because it distorts time with different ratios depending on the time window considered. There are fixed windows before (baseline) and after the word stimuli (waitpeak) where time should not be compressed/stretched. Also, the normalised time units are less interpretable than the absolute units in seconds, which becomes an issue in some applications of pupillometry.

In the current method, we simply minimized the effect of variable word duration by aligning the traces at the onset of the response before aggregating/averaging over repetitions, so that the time window of interest always captured the peak pupil response.

243-26 I like the comparison between the “block baseline” and the “word position baseline”.

Thank you!

252-291 I find it a little confusing (and quite demanding on my working memory) to present all statistical analyses prior to any results. I think this section would be helped by using sub-headings and, again, by providing the actual models in either lme4 syntax or as an equation. Alternatively, these paragraphs could be put as the initial paragraph of the respective results (sub)sections.

We thank the reviewer's advice on enhancing our delivery of the results. We have now re-organised the methods and results section so that readers could compare directly the results after reading the methods. This organisation is definitely more efficient and easier to digest.

265-282 I think the discussion of the second time window suggests that what you really should be doing is looking at the entire pupil diameter curve from the onset of the word-position to 1.5s after its offset. See Winn & Moore (2018) for a really clever way of breaking such long(ish) traces down for analysis.

Yes, their time series approach is interesting, but once again it really applies better to sentences, and when no memory task is involved and no verbal responses. Reviewer 1 asked a related question to determine the exact point at which the pupil traces separated. But with the current design – participants taking their time to report words with no time constraints, and the time window 2 contained verbal responses (see Limitation for more details) – it won’t be particularly useful.

Indeed, a curve analysis (growth curve analysis) supplied more information than the feature extraction analysis (extracting PPD, peak latency), with its rich information on the pupil size variation over time. But the feature extraction method still captured the most prominent task-evoked pupillary response, and with our trend analysis we could also gain information on its variation over time. While the slow cognitive related pupil variation (around 10Hz) had enough time to unravel in sentence stimuli, it could be tight in the duration of words.

The key point we wish to emphasize is that a larger PPD is not always a bad sign, provided that the task involved memory. The exact timing at which this differentiation happens is not particularly interesting and would likely only be determined with a design that places heavy constraints on the manner with which listeners are allowed to report the words recalled (which would itself hinder the memory task).

Figure 2 a minor point, but it seems needlessly complicated to present the results with different Y axes representing essentially the same thing - % correct words repeated vs. Average number of words remembered (presumably out of 10?)

Not at all: the % words repeated is about intelligibility, assessed when participants repeat back the word right after it was played. The number of stated words recalled is about memory, assessed at the end of a block. They are completely different DVs.

We have clarified this in the subtitle of the plots in the new figure 2 and its legend.

323-418 The results section is incredibly hard to read. Please revise to put things into complete sentences. It’s not just about presenting a bunch of equations here, you need to organize them in such a way that the reader can understand what you’re talking about. At this point I can’t really. Please give values (i.e. don’t just tell me baseline PD was bigger in one level than another, tell me what the value was for each level). It’s confusing to read that baseline pupil diameter was “bigger than 14 dB” (line 332) when to my knowledge we don’t typically measure pupil diameter in decibels. Yes, I can figure out what you mean, but this is currently written as it might be written in a lab notebook, for personal consumption, not as it should be written for scientific communication. And in some cases, it’s opaque: in line 335, is the (0.2 mm) referring to the absolute diameter, or the amount by which it is bigger? Even at the end “due to no corrections” is practically txting the results...

Also, I think the trend analyses could be discussed separately. Basically, right now it seems as if you're more or less just listing the results of all the tests you did, maybe in chronological order or perhaps loosely organized (?) according to dependent measure. Please consider some way of organizing the results in a way that facilitates the reader’s understanding of why you conclude what you will eventually conclude or, at a minimum, that reflects the issues that you determined were relevant to investigate as described in the introduction. Ideally, the results section should be presented in the same order as the discussion section, which should walk the reader through the

data toward the eventual theoretical claims that you want to make (and which should reflect the relative importance of topics as discussed in the introduction). Right now I honestly can’t figure out what data point(s) are particularly relevant or irrelevant, it just sort of devolved into a giant mass of statistical tests presented without obvious organization.

We thank the reviewer's feedback on the organisation of the section. We have addressed the issue by:

1) grouping the analysis method report with the results section directly for behavioural data, pupil data and subjective rating results.

2) adding meaningful subtitles for each chunk of analyses or results to indicate the purpose of the analysis.

3) better wording for more accurate number reporting.

[Discussion section is also a bit confusing - mostly due to digressions, though]

We have removed a few digressions to stay focussed on the organization outlined at the start of the discussion, namely 1) interference between concurrent tasks, 2) pupil dynamics, 3) predictive power of pupillometry, and 4) individual differences.

Figures 3 & 4 Looking at the traces in Figures 3a and 4a it seems apparent to me that peak pupil dilation may not

be a useful metric here. Except in the first word position there really isn’t much of a *peak* of any sort visible in 4a. And you can see that when those word positions get averaged together (for the images in 3a) any potential peakedness disappears. So why not use average PD or something like that? I think that would tell the story at least as well, and would be less subject to potentially weird micro-effects such as the weird flip of the black and red dots in positions 5, 8, and 9 of figure 4c.

That is fair point: average pupil dilation within a restricted window would likely do just as well, and the question becomes how narrow one should choose this window, and this would be certainly open to debate/criticism. By opting for the PPD, we took a more traditional approach that had the advantage of being directly comparable to methods used for sentences.

Also, the Y axis of 3c and 4c should somehow indicate that this is change from baseline.

Yes, we corrected them with “baseline-corrected PPD”.

In general, I’d recommend considering a very different way of doing this analysis, perhaps along the lines of Winn & Moore (2018).

As mentioned earlier, this might not apply directly to a memory task with single words.

340 should this be 3b or 4b?

Figure 4b shows the trend analysis from the 1st to 10th word.

432-443 references needed here to Pichora-Fuller et al. 1995, Surprenant (1999, 2007).

461-506 I really struggled with this discussion. I think the long and detailed references to the noise reduction work are distracting and superfluous. So lines 436-455 could be reduced to just lines 451-455.

We thank the reviewer for pointing out relevant literatures here! We have removed the effect of noise reduction here to only include studies relating to the recall performance. We also added in references to Pichora-Fuller et al., 1995 and Surprenant 1999.

Also, this discussion brings up the question of what, exactly, pupil dilation tells us. Arguably, it could provide information about the overall level of engagement of cognitive resources (I think that’s what the baseline measurement is supposed to get at, here) in these two conditions, as well as the moment-by-moment allocation of those resources during part of a task (encoding, repetition, recall). Given that you *have* pupil dilation data, I think this needs to be addressed somehow, before going into details of what dual task paradigms may or may not tell us.

And, finally, what do you conclude? I appreciate that there are multiple possible interpretations, but you've thought about this far more than most. Could you lead the reader from this apparent bafflement into something that we can be more satisfied with?

We thank the reviewer for pointing this out. We understand that although we have explored different possible interpretations, we didn’t do enough to provide a summarising or take-home points to ease the digestion of all the messages. Also, we didn’t do enough to connect the behavioural and pupillometry results to provide a linked picture.

To address this, we have emphasised the findings and the relevance of our experiment at the end of both the behavioural (line567) and pupillary results (line682, line752). WE have also strengthened the link between our behavioural and pupillary findings (line520) .

488 Could you examine age differences in your data? What would you predict to see either in terms of behavior or pupil dilation if people are prioritizing things differently?

We further tested and discussed the effect of age on the individual performances in the later section titled Individual Differences, also some hypothesis to test at line694. There are however no easy answers to the question on prioritizing tasks. One could image that someone who prioritizes the repeat task would show a smaller PPD (and better intelligibility), along with a lower baseline (and worse recall). But given the observed saturation of the PPD, we would speculate that a listener prioritizing the recall task is also likely to exhibit a small PPD (and poorer intelligibility), along with a higher baseline (and better recall). Whether age would incite the former or the latter pattern is itself largely unclear.

490 what does it mean that the recall paradigm is from previous studies? Which recall paradigm?

We have added the references to the two recall paradigms mentioned earlier (line553).

498 Define SWIR acronym.

Clarified here by referring to the references (line562).

509 What second hypothesis? There are so many hypotheses swirling around by now I’ve lost track of which one is which. Please restate.

Clarified.

519 these references did not all use the same speech perception task. Clarify.

We have re-organised the references to be clear (line589).

523 You don’t really have data showing any greater effort of your task over other tasks.

Indeed a direct comparison is not available here. We have changed the wording (line592).

540 The lack of position effect is extremely unusual for a serial recall task and needs to be discussed in much more detail. It should also be presented in the results. This is one of those memory effects that is so basic it’s taught in intro psych textbooks... I would very much like to see a graph of word recognition and recall by word position. I have great difficulty imaging that there wasn’t some kind of recency effect at least, if not also a primacy effect, with a 10-item list to be recalled.

Certainly, we replicate the recency and primacy effects just as expected. We had not reported these results in the main body because we did not observe an interaction with SNR conditions, therefore, it was not directly tied to the main purpose of the experiment. But this is surely a finding that would reassure the reader, so we have decided to include them in a third Supplementary material.

585-621 It seems to me that the best explanation for smaller growth of the PPD is that the baseline is increasing.

So the limit (probably physiological, based on light levels) is imposed not in terms of how much the pupil can dilate, but in terms of how much of a dilation it will reach. In other words, illumination, which you held constant, may have imposed an upper limit on pupil dilation, such that as the baseline creeps up with increasing memory load in the recall condition, or creeps down with increasing habituation in the repeat-only condition, you get the difference between the two gradually shrinking (in the recall condition) or increasing (in the repeat only condition). No, illumination did not impose an upper limit on pupil dilation. We know this because the pupil did jump considerably as soon as participants were prompted to report as many words as possible. This is a huge effect compared to the PPD measured by the end of the block (when it seems like it’s saturating), meaning that there was

definitely space for the pupil to dilate more. This phenomenon seems to have been overlooked by Reviewer 1 as well, so we highlighted this effect more explicitly (line682). In other words, the apparent saturation was of cognitive origin, not mechanical. Also, refer to our response to the first comment by Reviewer 1 on the same idea.

623-626 Word choice seems problematic. What does it mean to “hold predictive power” or to be “responsive for recall performance”? Say what you want to say in a simple way.

We have simplified wording.

References that should be incorporated into a revision

Goldinger, S. D., & Papesh, M. H. (2012). Pupil dilation reflects the creation and retrieval of memories. Current directions in psychological science, 21(2), 90-95.

Kucewicz, M. T., Dolezal, J., Kremen, V., Berry, B. M., Miller, L. R., Magee, A. L., ... & Worrell, G. A. (2018). Pupil size reflects successful encoding and recall of memory in humans. Scientific reports, 8(1), 1-7.

Miller, A. L., Gross, M. P., & Unsworth, N. (2019). Individual differences in working memory capacity and long-term memory: The influence of intensity of attention to items at encoding as measured by pupil dilation. Journal of Memory and Language, 104, 25-42.

Pichora-Fuller, M. K., Schneider, B. A., & Daneman, M. (1995). How young and old adults listen to and remember speech in noise. The Journal of the Acoustical Society of America, 97(1), 593-608.

Surprenant, A. M. (1999). The effect of noise on memory for spoken syllables. International Journal of Psychology, 34(5-6), 328-333.

Surprenant, A. M. (2007). Effects of noise on identification and serial recall of nonsense syllables in older and younger adults. Aging, Neuropsychology, and Cognition, 14(2), 126-143.

Winn, M. B., & Moore, A. N. (2018). Pupillometry reveals that context benefit in speech perception can be disrupted by later-occurring sounds, especially in listeners with cochlear implants. Trends in hearing, 22, 2331216518808962

Thank you for these references, which we included.

---

## [Decision Letter · Decision Letter 1]

19 Oct 2020

PONE-D-20-12674R1

Disentangling listening effort and memory load beyond behavioural evidence:

Pupillary response to listening effort during a concurrent memory task

PLOS ONE

Dear Dr. Zhang,

Thank you for submitting your manuscript to PLOS ONE. Your revised manuscript has been reviewed by the original reviewer. One is satisfied with your revision while the other request further clarification. After careful consideration, we feel that it has merit but does not fully meet PLOS ONE’s publication criteria as it currently stands. Therefore, we invite you to submit a revised version of the manuscript that addresses the points raised during the review process.

We look forward to receiving your revised manuscript.

Kind regards,

Claude Alain

Academic Editor

PLOS ONE

Reviewers' comments:

Reviewer's Responses to Questions

**Comments to the Author**

1. If the authors have adequately addressed your comments raised in a previous round of review and you feel that this manuscript is now acceptable for publication, you may indicate that here to bypass the “Comments to the Author” section, enter your conflict of interest statement in the “Confidential to Editor” section, and submit your "Accept" recommendation.

Reviewer #1: (No Response)

Reviewer #2: All comments have been addressed

2. Is the manuscript technically sound, and do the data support the conclusions?

Reviewer #1: Partly

Reviewer #2: Yes

3. Has the statistical analysis been performed appropriately and rigorously? 

Reviewer #1: No

Reviewer #2: Yes

4. Have the authors made all data underlying the findings in their manuscript fully available?

Reviewer #1: Yes

Reviewer #2: Yes

5. Is the manuscript presented in an intelligible fashion and written in standard English?

Reviewer #1: Yes

Reviewer #2: Yes

6. Review Comments to the Author

Reviewer #1: The revised version of the manuscript gained on clarity and furthermore improved in its readability. I have read the authors’ responses and I am happy with most of them. However, two of my major concerns —which are also actually the most important major concerns of my first review— still require authors’ response.

I also checked the raw pupil data the authors uploaded. It’s great, however, it’s unclear if each row represents a time sample. To make the analysis reproducible, could you please also add the time coordinate for each entry?

——————

# MAJOR CONCERN #1:

## REVIEWER 1 IN ROUND 1:

1. The possibly most important finding of this study is the fact the PPD (peak pupil diameter) tends to be smaller in the last few words in the repeat-to-recall condition. The authors were very surprised by this result and tried to interpret it by comparing with the previous listening effort studies like Zekveld et al, 2019. However, the explanation the authors offered in the discussion was extensive but not satisfying. It has been well-known that PPD is not only related to the effort or load but also strongly related to its baseline; the larger the baseline, the smaller the PPD. Figure 4b clearly showed that the large baseline is the case. Thus, a simple explanation for this result is that pupil simply saturated in the repeat with recall condition and the pupil simply cannot expand further in the presence of additional words and responses. If this is the case, the result is not surprising at all.

To exclude this possibility, the authors should consider running further analysis (e.g. regress out the effect of baseline from PPD) or conducting additional experiments to show that pupil still CAN dilate further in the repeat with recall condition. If these cannot be done, the authors should at least discuss it in the discussion. The saturation could be not only due to the mechanical limitation of the muscles controlling the pupil diameter but also because pupil diameter is strongly correlated with the norepinephrine activity in the LC system. Since the authors are aware of the link between pupil diameter and LC-NE system as this was briefly mentioned in Introduction (line 36), they should also take this into account in the discussion.

## Authors:

We thank the reviewer’s contribution to this interpretation of the results.

We agree that baseline and PPD are correlated mathematically because (X-Y) and Y will always be correlated by R2=0.5, assuming X and Y completely random. And past literatures have also demonstrated this correlation. However, the exact relation between baseline and PPD during a hearing or cognitive task depends on the underlying cause. For instance, the effect of old age induces smaller baseline and smaller PPD due to physiological constraints and changes of activity in peripheral and/or central nervous system (Piquado et al., 2010; Kuchinsky et al., 2016; Wang et al., 2018). Lower luminance induces bigger baseline but smaller PPD due to the ‘gripping’ of parasympathetic system (Wang et al., 2018). Therefore, it is unclear what direction this relation between PPD and baseline should be in a task with concurrent listening and cognitive demands. What is surprising in our results is that we initially hypothesised that PPD might increase with more difficult SNR and more items to retain in the memory, but we see instead that the pupil dilation ‘capped’ during the listening section (before the word recall section). However, looking at the baseline dynamics let us understand partially the cause of the ‘capping’. This highlights the importance to look both at the PPD and baseline in future experiments that involves more ecologically realistic tests.

We share with Reviewer 1 the desire to further disentangle baseline from PPD by either regressing out the effect

of baseline or showing that pupil can still dilate further in repeat with recall condition. The first approach, however, is problematic as long as we do not understand the exact conditions where base and PPD are negatively correlated from conditions where they may be positively correlated (whether this is seen within or across subjects). So, we opted for the second approach: while a sort of pupil saturation was present during the listening and encoding section from 1st to 10th word, the pupil increased at the onset of recall on average by 0.3mm! Reviewer 1 did not realize this finding, so we made it more explicit in the article: the ceiling of the pupil during the recall blocks cannot be due to mechanical limitation of the muscles controlling the pupil diameter, because right at the end of the block, the pupil diameter rose considerably, an effect equivalent to six times the average PPD at the 10th word. Therefore, it is clear that the pupil ceiling during listening and encoding was not at all due to mechanical constraints but originated from cognitive resource allocation strategy. The best interpretation we can offer – and that we discussed – is that listeners would reserve their resources during the 1st to 10th word in order to retrieve the words during the recall section.

## REVIEWER 1 IN ROUND 2:

Sorry for being very fussy about this. This has been surprisingly under-addressed in the literature. As you are aware of it now, please properly discuss it in the manuscript and point out that, although it’s less exciting, it is a reasonable explanation of your result.

(1) I am not convinced by the authors' rejection to regress out the baseline from PPD on a trial basis. As shown in the first paragraph of their response, the authors clearly understood the concern about the correlation between baseline and PPD. As such correlation potentially exist and explains the key result, it should be carefully examined and reported, because it potentially “fully” not just “partially” explains the key finding here. No matter it’s a positive or negative correlation, no matter it’s within- or across- subjects.

(2) The authors’ response to my second approach is not satisfying either. Remember, the key finding here is the PPD (peak pupil diameter) tends to be smaller in the last few words in the repeat-to-recall condition. The authors need to examine if this is due to the pupillary saturation in the last few words. In other words, the authors need to show that in the last few words, the pupil can still dilate more flexibly just like in other conditions. The dilation from 1st to 10th word (as shown in figure 4b) does not solve the concern at all. Actually, based on figure 4b, the baseline reached a plateau between 4.0-4.1mm after the 6th word, strongly suggesting that the saturation is the case.

(3) Moreover, I am not convinced by the authors’ statement that “The best interpretation we can offer – and that we discussed – is that listeners would reserve their resources during the 1st to 10th word to retrieve the words during the recall section.” This statement is strong, but the link between “small PPD in the last few words” and “reserving the resource” is weak to me. Please elaborate on it.

——————

# MAJOR CONCERN #2

## REVIEWER 1 IN ROUND 2:

5. [line 240] How was PPD computed here? Was it extracted from each trial and then averaged within each subject? Or was PPD directly extracted from each subject’s average pupil diameter response?

## AUTHORS: We thank Reviewer 1 for this comment, which we have repeatedly heard while presenting these results at conferences. It is a matter of constant debate. For sentences, the pupil dilation is more stable than it is for individual words, and thus arguably, one might want to extract PPD directly from individual sentences. We found that this did not apply well to individual words. So, we opted for the PPD taken from the averaged traces. We

firstly performed the baseline correction to subtract the baseline of each trial from the pupil trace. Then traces were aligned by the onset of the response prompt and aggregated per listener per condition. PPD was then calculated at this aggregated level, instead of the trial level. This method was chosen in aligned with past studies and ensured PPD was more robust (Zekveld et al., 2010; Zekveld et al., 2013; Zekveld et al., 2014).

We have re-organised the method and result sections to clarify the detailed procedure (line280).

## REVIEWER 1 IN ROUND 2:

Thanks.

(1) could you clarify what you meant by “this did not apply well to individual words”? How did you determine that method doesn’t apply well?

(2) although it’s ok to choose this method as it has been consistently used by Zekveld lab since 2010, it does not mean that in 2020 we, the pupillometry field, should still ONLY reply on this simple method. So, to demonstrate that your result is robust and replicable, please at least report the result with the trial level PPD in supplementary materials.

Reviewer #2: (No Response)

7. PLOS authors have the option to publish the peer review history of their article (what does this mean?). If published, this will include your full peer review and any attached files.

Reviewer #1: **Yes: **Sijia Zhao

Reviewer #2: **Yes: **Alexander L. Francis

---

## [Author Response · Author response to Decision Letter 1]

3 Dec 2020

Reviewer #1: The revised version of the manuscript gained on clarity and furthermore improved in its readability. I have read the authors’ responses and I am happy with most of them. However, two of my major concerns —which are also actually the most important major concerns of my first review— still require authors’ response.  I also checked the raw pupil data the authors uploaded. It’s great, however, it’s unclear if each row represents a time sample. To make the analysis reproducible, could you please also add the time coordinate for each entry?

We have now uploaded pupil data files with extra columns representing aligned sequence in time. They are aligned by the onset of baseline (basAtime), onset of word (wordAtime), onset of word repetition (wordAtime), onset of recall (speakAtime).

 ## REVIEWER 1 IN ROUND 2: Sorry for being very fussy about this. This has been surprisingly under-addressed in the literature. As you are aware of it now, please properly discuss it in the manuscript and point out that, although it’s less exciting, it is a reasonable explanation of your result.

We must not have been clear enough in our first response. It is simply not true that the pupil saturated in the last few words of the lists in the repeat-with-recall condition. And we know that because the pupil jumped by 0.3 mm as soon as the recall started. Please refer to Figure 6 showing the sudden jump which proves that the baseline at the final few trials may be high (around 4 mm), it does NOT prevent the pupil to dilate further. In this regard, we do not need any further experiments to demonstrate that this is case because we have done it: it is shown in Figure 6. Please also find the attached figure to show the pupil trace 15s before the onset of recall. Although every 10-words block varied in time due to different response time, on average 15s covered around the last 5 words in the list. Please notice how, regardless of how high the baseline and PPD already were in the final few seconds during the word listening, pupil size still increased when the recall stage started. It is clear that there is no physical constraints on pupil dilation and the size could increase at the end of the repeat-with-recall block.

(1) I am not convinced by the authors' rejection to regress out the baseline from PPD on a trial basis. As shown in the first paragraph of their response, the authors clearly understood the concern about the correlation between baseline and PPD. As such correlation potentially exist and explains the key result, it should be carefully examined and reported, because it potentially “fully” not just “partially” explains the key finding here. No matter it’s a positive or negative correlation, no matter it’s within- or across- subjects.

The inherent mathematical relationship between PPD (y-x) and baseline (x) is a theoretical relationship. Even with two variables, x and y, that are completely random and independent of one another, x shares 50% of the variance with x-y. Yet, in practice, there are many factors that can exacerbate this link (i.e. make its r2 stronger than 0.5) or on the contrary impair this link (i.e. make its r2 weaker than 0.5). But we know that this inherent relationship does NOT fully explain the capping of the pupil towards the end of the list, because as soon as recall starts the pupil (which is at a high baseline, 4 mm) can dilate to a pretty impressive amount, by 0.3 mm. This is twice the PDD observed by the end of the list. Therefore, evidently, the pupil CAN dilate even when the baseline is high. We need no further evidence that the pupil was not mechanically constrained by this baseline being at 4mm. 

 (2) The authors’ response to my second approach is not satisfying either. Remember, the key finding here is the PPD (peak pupil diameter) tends to be smaller in the last few words in the repeat-to-recall condition. The authors need to examine if this is due to the pupillary saturation in the last few words. In other words, the authors need to show that in the last few words, the pupil can still dilate more flexibly just like in other conditions. The dilation from 1st to 10th word (as shown in figure 4b) does not solve the concern at all. Actually, based on figure 4b, the baseline reached a plateau between 4.0-4.1mm after the 6th word, strongly suggesting that the saturation is the case.

We would argue that one of the key finding is also the large jump (which is twice the PPD size) that occurs after the 10th word, while the baseline is at 4 mm. So, figure 4b is not the evidence you are looking for; figure 6 is which demonstrates that there is NO saturation. In other words, the plateau that is visible during the 10 words is not of physical or mathematical limit, but of cognitive origin. This is what we discussed extensively (line 638). And we also cited past study showing similar pupillary response for digit recall tasks (line654). 

 (3) Moreover, I am not convinced by the authors’ statement that “The best interpretation we can offer – and that we discussed – is that listeners would reserve their resources during the 1st to 10th word to retrieve the words during the recall section.” This statement is strong, but the link between “small PPD in the last few words” and “reserving the resource” is weak to me. Please elaborate on it.

We are turning around the same point: as the limitation comes from a cognitive source, there must be cognitive reasons why listeners do not dilate more during the listening-and-repeat task, and instead reallocate their resources to the task coming after the 10th word. We elaborated extensively on this topic from line 650 onwards. —————— # MAJOR CONCERN #2  ## REVIEWER 1 IN ROUND 2: Thanks. (1) could you clarify what you meant by “this did not apply well to individual words”? How did you determine that method doesn’t apply well?

Very simply, words are short in comparison with sentences, on the order of 700 ms versus 3-4 seconds. As you know, there is intrinsic variability in the pupil dynamics over time. With a longer auditory event (such as a sentence), it is more likely that the pupil will dilate at some point over the next 5 seconds. There may be occasional traces that show a negative peak but they are rare. This is not rare with individual words, as the auditory event is very short and places little demand on the listener (especially here with normally-hearing and relatively young adults). Kuchinsky et al. (2014) used individual words and applied a similar pre-processing method as here, i.e. aggregating across all trials for a given listener and condition. It’s not impossible to analyze the data on a trial-by-trial basis but it implies a good deal of variability in the shape of the pupil response, and any naïve reader may feel skeptical to grant meaning to a PPD when the shape of the pupil trace is largely flat or even going down. So, we simply followed recommended practice (Winn et al., 2018). 

 (2) although it’s ok to choose this method as it has been consistently used by Zekveld lab since 2010, it does not mean that in 2020 we, the pupillometry field, should still ONLY reply on this simple method. So, to demonstrate that your result is robust and replicable, please at least report the result with the trial level PPD in supplementary materials.

Of course, not because something has been done in a certain way in the past implies that we should follow it. But here, we very much agree with past recommendations and there are strong reasons why to do so, namely with regard to the stereotypical shape of the pupil trace during an auditory event. In the literature, there are many studies that first average across all trials of a block, and then calculate baseline and PPD. This is NOT what we have done because we believe there is much information to be gained by looking at the evolution of the pupil dynamics within a block (and have shown many pieces of evidence throughout our manuscript to prove it). We calculated baseline and PPD for each position in the block, but since we had three repetitions of each experimental condition, this allowed us to aggregate to a very small degree the three traces of each position, before deriving our pupillary metrics. In contrast to what Review 1 is asking, we have been asked several times through conferences to show the opposite, i.e. first aggregate the traces through all repetitions of a condition (in our case the 30 trials) and derive baseline and PPD from the average traces. We have done so and shown additional analysis methods in the supplementary materials already. We think that it would be confusing to add yet another analysis method that would differ very little from the present one.

---

## [Editor Report · Decision Letter 2]

16 Feb 2021

Disentangling listening effort and memory load beyond behavioural evidence:

Pupillary response to listening effort during a concurrent memory task

PONE-D-20-12674R2

Dear Dr. Zhang,

Thank you for the revision of your manuscript and for your patience in awaiting our response. For some reason, I could not reach one of the reviewers anymore, although s/he agreed to review the manuscript. I now decided to not wait any longer. We’re pleased to inform you that your manuscript has been judged scientifically suitable for publication and will be formally accepted for publication once it meets all outstanding technical requirements.

Kind regards,

Claude Alain

Academic Editor

PLOS ONE

Additional Editor Comments (optional):

Thank you for the revision of your manuscript and for your patience in awaiting our response. For some reason, I could not reach one of the reviewers anymore. I now decided to not wait any longer and am pleased to tell you that your work has now been accepted for publication in PLoS ONE.
---

## [Editor Report · Acceptance letter]

22 Feb 2021

PONE-D-20-12674R2 

Disentangling listening effort and memory load beyond behavioural evidence:
Pupillary response to listening effort during a concurrent memory task 

Dear Dr. Zhang:

I'm pleased to inform you that your manuscript has been deemed suitable for publication in PLOS ONE. Congratulations! Your manuscript is now with our production department. 

Kind regards, 

on behalf of

Dr. Claude Alain 

Academic Editor

PLOS ONE